# The DeepMIP contribution to PMIP4: experimental design for model simulations of the EECO, PETM, and pre-PETM (version 1.0)

Daniel J. Lunt[1], Matthew Huber[2], Eleni Anagnostou[9], Michiel L.J. Baatsen[3], Rodrigo Caballero[4], Rob DeConto[5], Henk A. Dijkstra[3], Yannick Donnadieu[6], David Evans[31], Ran Feng[8], Gavin L. Foster[9], Ed Gasson[5], Anna S. von der Heydt[3], Chris J. Hollis[10], Gordon N. Inglis[32], Stephen M. Jones[34], Jeff Kiehl[29], Sandy Kirtland Turner[11], Robert L. Korty[12], Reinhardt Kozdon[13], Srinath Krishnan[7], Jean-Baptiste Ladant[6], Petra Langebroek[14], Caroline H. Lear[15], Allegra N. LeGrande[16], Kate Littler[17], Paul Markwick[18], Bette Otto-Bliesner[8], Paul Pearson[15], Christopher J. Poulsen[19], Ulrich Salzmann[20], Christine Shields[8], Kathryn Snell[21], Michael Starz[22], James Super[7], Clay Tabor[8], Jess Tierney[23], Gregory J.L. Tourte[1], Aradhna Tripati[33], Gary R. Upchurch[24], Bridget S. Wade[25], Scott L. Wing[26], Arne M.E. Winguth[27], Nicky Wright[28], James C. Zachos[29], and Richard Zeebe[30]

[1]School of Geographical Sciences, University of Bristol, UK
[2]Department of Earth Sciences, University of New Hampshire, USA
[3]Institute for Marine and Atmospheric research Utrecht (IMAU), Utrecht University, Netherlands
[4]Department of Meteorology (MISU), Stockholm University, Sweden
[5]Department of Geosciences, University of Massachusetts-Amherst ,USA
[6]Laboratoire des Sciences du Climat et de l'Environnement, CNRS/CEA, France
[7]Department of Geology and Geophysics, Yale University, USA
[8]National Centre for Atmospheric Research, USA
[9]Ocean and Earth Science, National Oceanography Centre Southampton, University of Southampton, UK
[10]GNS Science, New Zealand
[11]Department of Earth Sciences, University of California - Riverside, USA
[12]Department of Atmospheric Sciences, Texas A&M University, USA
[13]Lamont-Doherty Earth Observatory of Columbia University, USA
[14]Uni Research Climate, Bjerknes Centre for Climate Research, Norway
[15]School of Earth and Ocean Sciences, Cardiff University
[16]NASA-GISS, USA
[17]Camborne School of Mines, University of Exeter, UK
[18]Getech Group plc, UK
[19]Department of Earth and Environmental Sciences, University of Michigan, USA
[20]Department of Geography, Northumbria University, UK
[21]Department of Geological Sciences, University of Colorado, USA
[22]Alfred Wegener Institute, Germany
[23]Department of Geosciences, University of Arizona, USA
[24]Department of Biology, Texas State Univesity, USA
[25]Department of Earth Sciences, University College London, UK
[26]Department of Paleobiology, Smithsonian Institution, USA
[27]Earth and Environmental Science, University of Texas - Arlington, USA
[28]School of Geosciences, University of Sydney, Australia
[29]PBSci-Earth & Planetary Sciences Department, Institute of Marine Sciences, University of California - Santa Cruz, USA
[30]Department of Oceanography, University of Hawaii at Manoa, USA

[31]Earth Sciences, University of St Andrews, UK

[32]School of Chemistry, University of Bristol, UK

[33]Earth, Planetary, and Space Sciences, Atmospheric and Oceanic Sciences, Institute of the Environment and Sustainability, University of California - Los Angeles, USA

[34]School of Geography, Earth and Environmental Sciences, University of Birmingham, UK

*Correspondence to:* Dan Lunt (d.j.lunt@bristol.ac.uk)

**Abstract.** Past warm periods provide an opportunity to evaluate climate models under extreme forcing scenarios, in particular high ($>800\,\mathrm{ppmv}$) atmospheric $CO_2$ concentrations. Although a post-hoc intercomparison of Eocene ($\sim$50 million years ago, $\mathrm{Ma}$) climate model simulations and geological data has been carried out previously, models of past high-$CO_2$ periods have never been evaluated in a consistent framework. Here, we present an experimental design for climate model simulations

of three warm periods within the early Eocene and the latest Paleocene (the EECO, PETM, and pre-PETM). Together with the CMIP6 preindustrial control and abrupt $4\times CO_2$ simulations, and additional sensitivity studies, these form the first phase of DeepMIP – the Deep-time Model Intercomparison Project, itself a group within the wider Paleoclimate Modelling Intercomparison Project (PMIP). The experimental design specifies and provides guidance on boundary conditions associated with palaeogeography, greenhouse gases, astronomical configuration, solar constant, land surface processes, and aerosols. Initial

conditions, simulation length, and output variables are also specified. Finally, we explain how the geological datasets, which will be used to evaluate the simulations, will be developed.

## 1   Introduction

There is a large community of Earth scientists with strong interests in 'deep-time' palaeoclimates, here defined as climates of the pre-Pliocene (i.e., prior to $\sim$5 Ma). Recently, a growing community of modelling groups focussing on these periods

is also beginning to emerge. DeepMIP – the Deep-time Model Intercomparison Project – brings together modellers, the data community, and other scientists, into a multidisciplinary international effort dedicated to conceiving, designing, carrying out, analysing, and disseminating an improved understanding of these time periods. It also aims to assess their relevance for our understanding of future climate change. DeepMIP is a working group in the wider Paleoclimate Modelling Intercomparison Project (PMIP4), which itself is a part of the sixth phase of the Coupled Model Intercomparison Project (CMIP6, Eyring

et al., 2016). In DeepMIP, we will focus on three time periods in the latest Paleocene and early Eocene ($\sim$55–50 Ma), and for the first time, carry out a formal coordinated model–data intercomparison. In addition to the experimental design presented here, DeepMIP will synthesise existing climate proxy records, and develop new ones if appropriate. The aim will be to best characterise our understanding of the palaeoclimate of the chosen interval through the synthesis of climate proxy records, to compare this with the model simulations, and to understand the reasons for the intra and inter model and data differences. The

ultimate aim is to encourage model development in response to any robust model deficiencies that emerge from the model–data comparison. This is of particular relevance to models that are also used for future climate projection, given the relative warmth and high $CO_2$ that characterises many intervals of deep-time.

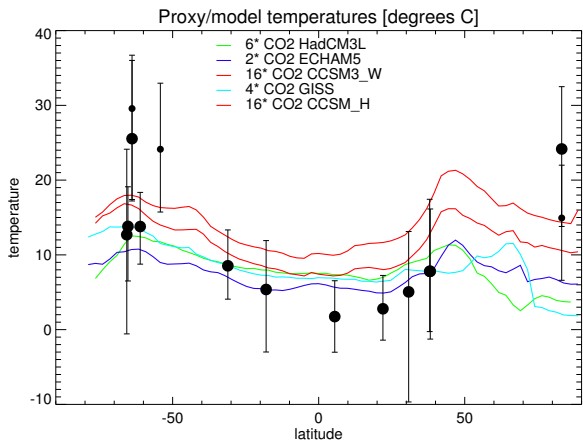

**Figure 1.** Zonal mean Eocene sea surface temperature warming, presented as an anomaly relative to present/pre-industrial. Warming from the five models in 'Eomip' (Lunt et al., 2012) are shown as coloured lines; for each model only the $CO_2$ concentration that best fits the temperature proxy observations is shown. Warming derived from the proxies are shown as filled circles, with error bars representing the range of uncertainty associated with proxy calibration and temporal variability. Larger symbols represent 'background' early Eocene state, smaller symbols represent the EECO. Adapted from Figure 8a in Lunt et al. (2012).

## 2 Previous Work

An informal, post-hoc model–data intercomparison has previously been carried out for the early Eocene (Lunt et al., 2012). This compared the results of four models from five modelling groups with marine and terrestrial data syntheses, and explored the reasons for the model–model differences using energy balance diagnostics. That study contributed to the recent IPCC AR5
5   report (Box 5.1, Fig. 1), but it also revealed challenging differences between model simulations of this period, intriguing model–data mismatches, as well as inconsistencies between proxies (Figure 1). For example, proxy-derived SST estimates indicate a weak meridional temperature gradient during the early Eocene which cannot easily be reconciled with the model simulations. Further work resulting from this intercomparison included Gasson et al. (2014), which investigated the $CO_2$ thresholds for Antarctic ice sheet inception; Lunt et al. (2013), which compared the ensemble and data to further Eocene simulations; and
10  Carmichael et al. (2016), which investigated the hydrological cycle across the ensemble and compared model results with proxies for precipitation.

The previous exercise points to the need for a more coordinated experimental design (different modelling groups had carried out simulations with different boundary conditions, and different initial conditions etc.), and a greater understanding for the reasons behind differences between different climate proxies. Those challenges provide the motivation for DeepMIP.

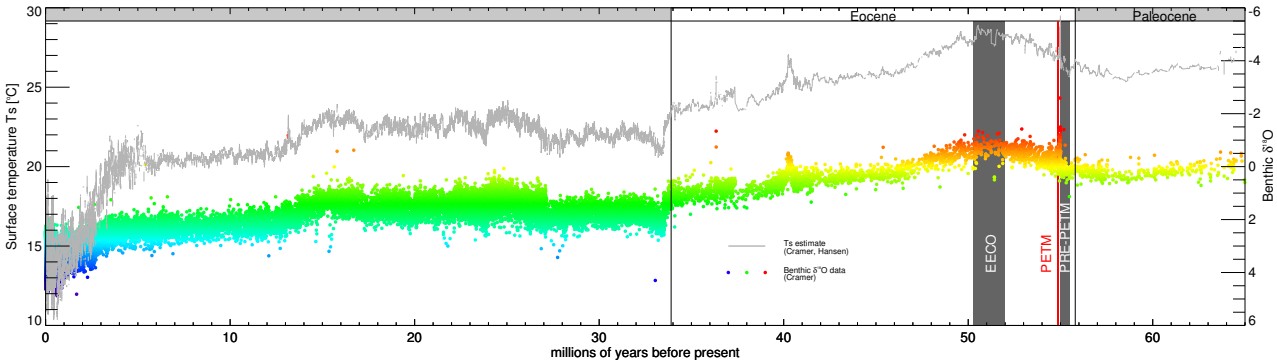

**Figure 2.** The three DeepMIP palaeo intervals - EECO (grey shaded region), pre-PETM (grey shaded region), and the PETM (vertical red line). Also shown for context is the climate evolution over the last 65 million years, as expressed by the benthic oxygen isotope record of Cramer et al. (2009) (coloured dots), and a surface temperature record produced by applying the methodologies of Hansen et al. (2013) to the Cramer et al. (2009) $\delta^{18}O_{\mathrm{benthic}}$ data, and applying a 10-point running average (grey line). Note that the formal definition of the start and end date of each time period is still to be finalised.

## 3   The chosen intervals – the Early Eocene Climatic Optimum (EECO) the Paleocene–Eocene Thermal Maximum (PETM), and the pre-PETM.

The choice of time interval on which to focus is based on a balance between (i) the magnitude of the anticipated climate
signal (larger signals have a higher signal-to-uncertainty ratio, and larger signals provide a greater challenge to models), (ii) the uncertainties in boundary conditions that characterise the interval (small uncertainties result in more robust conclusions as to the models' abilities, and minimise the model sensitivity studies required to explore the uncertainties), and (iii) the amount and geographic distribution of palaeoclimate data available with which to evaluate the model simulations.

We have chosen to focus on the latest Paleocene and early Eocene – $\sim$55 to $\sim$50 Ma (the Ypresian stage), as it is the most
recent geological interval characterised by high ($>$800 ppmv) atmospheric $CO_2$ concentrations. Within the latest Paleocene and early Eocene, DeepMIP will focus on three periods (see Figure 2):

1. The Early Eocene Climatic Optimum (EECO, $\sim$53–51 Ma)
   which is the period of greatest sustained ($>$1 Myr) warmth in the last 65 million years.

2. The Paleocene-Eocene Thermal Maximum (PETM, $\sim$55 Ma)
which is the event of greatest warmth in the last 65 million years.

3. The period just before the PETM (pre-PETM, or latest Paleocene)
   which is relatively warm compared with modern, but is cooler than both the PETM and the EECO.

These intervals have been the focus of numerous studies in the geological literature, and some syntheses of proxies from these intervals already exist (e.g. Huber and Caballero, 2011; Lunt et al., 2012; Dunkley Jones et al., 2013). The pre-PETM

**Table 1.** Summary of simulations associated with DeepMIP, including two relevant simulations from CMIP6 (*piControl* and *abrupt-4×CO2*), the three standard simulations (*deepmip-stand-X*), and some of the suggested sensitivity studies (*deepmip-sens-X*).

| Simulation Name | Simulation description | $CO_2$ [ppmv] | palaeogeography |
|---|---|---|---|
| *piControl* | preindustrial control (Eyring et al., 2016) | 280[1] | modern |
| *abrupt-4×CO2* | abrupt increase to $4\times$ $CO_2$ concentrations (Eyring et al., 2016) | 1120 | modern |
| *deepmip-stand-3×CO2* | pre-PETM, at $3\times$preindustrial $CO_2$ | 840 | Herold et al. (2014) |
| *deepmip-stand-6×CO2* | EECO/PETM, at $6\times$preindustrial $CO_2$ | 1680 | Herold et al. (2014) |
| *deepmip-stand-12×CO2* | EECO/PETM, at $12\times$preindustrial $CO_2$ | 3360 | Herold et al. (2014) |
| *deepmip-sens-Y×CO2* | Sensitivity study at $Y\times$preindustrial $CO_2$ | $Y\times 280$ | Herold et al. (2014) |
| *deepmip-sens-geoggetech* | Sensitivity study with modified palaeogeography | 840, 1680, 3360[2] | Lunt et al. (2016) |
| *deepmip-sens-geogpalmag* | Sensitivity study with modified palaeogeography | 840, 1680, 3360[2] | *This paper* |

[1] If a value different from 280 ppmv is used for $piControl$, then all other $CO_2$ values in the table should be changed accordingly.

[2] Order of priority, highest priority first.

provides a reference point for both the PETM and the EECO. In addition, all three time periods can be referenced to modern or pre-industrial. This is in recognition that both modelling and proxies are most robust when considering relative changes, as opposed to absolutes.

5     Compared to earlier warm periods, such as the mid-Cretaceous, the palaeogeography during the early Eocene is reasonably well constrained, and freely available digital palaeogeographic datasets exist; however, there are wide uncertainties in estimates of atmospheric $CO_2$ at this time. Furthermore, due at least in part to interest in the Eocene and PETM for providing information of relevance to the future (e.g. Anagnostou et al., 2016; Zeebe et al., 2016), there is a relative wealth of climate proxy data with which the model results can be compared.

## 4   Experimental design

The DeepMIP experimental protocol consists of five main simulations - pre-industrial, future, two in the early Eocene (EECO and PETM), and one in the latest Paleocene (pre-PETM), plus a number of optional sensitivity studies (see Section 4.3). The simulations are summarised in Table 1.

### 4.1   Pre-industrial and future simulations

15 The pre-industrial simulation should be as close as possible to the CMIP6 standard, *piControl* (Eyring et al., 2016). Many groups will already have carried out this simulation as part of CMIP6. Some groups may need to make changes to their CMIP6 model configuration for the DeepMIP palaeoclimate simulations (for example changes to ocean diffusivity). If this is the case,

we encourage groups to carry out a new preindustrial simulation with the model configuration used for DeepMIP palaeoclimate simulations.

The future simulation is the CMIP6 standard *abrupt-4×CO2* simulation (Eyring et al., 2016), which branches off from the *piControl* simulation, and in which atmospheric $CO_2$ is abruptly quadrupled and then held constant for at least 150 years.

## 4.2   EECO/PETM and pre-PETM simulations

This section describes the DeepMIP palaeoclimate simulations. There are three standard palaeoclimate simulations (*deepmip-stand-3×CO2*, *deepmip-stand-6×CO2*, *deepmip-stand-12×CO2*), which differ only in their atmospheric $CO_2$ concentration, plus a number of optional sensitivity studies. In general terms, we consider the *deepmip-stand-3×CO2* simulation as representative of the pre-PETM, and the other two simulations as representing two different scenarios for the EECO and/or PETM.

### 4.2.1   Palaeogeography and land-sea mask

Herold et al. (2014, henceforth H14) is a peer-reviewed, traceable, freely-available digital reconstruction of the early Eocene interval. It includes topography and sub-gridscale topography, bathymetry, tidal dissipation, vegetation, aerosol distributions, and river runoff. The palaeogeography from H14 should be used for all the standard DeepMIP palaeoclimate simulations (see Table 1); they are provided digitally in netcdf format in the Supplementary Information of H14 (see Table 2), at a resolution of $1° × 1°$, and are illustrated here in Fig. 3(a). The palaeogeographic height should be applied as an absolute, rather than as an anomaly to the pre-industrial topography. Most models additionally require some fields related to the subgridscale orography to be provided. Because subgridscale orographies are very sensitive to the resolution of the underlying dataset, the subgridscale orography (if it is required by the model) can be estimated based on fields also provided in Supplementary Information of H14. This can be implemented as the modelling groups see fit, but care should be taken that the pre-industrial and Eocene subgridscale topographies are as consistent as possible. In addition, the code used to calculate the subgridscale orographies in the CESM (Gent et al., 2011) model is also provided in the Supplementary Information of H14.

The land-sea mask can be initially calculated from the palaeogeographic height, by assigning ocean to palaeogeographic heights less than or equal to zero. Care should be taken when defining the land-sea mask for the ocean component of the model that the various seaways are preserved at the model resolution; this may require some manual manipulation of the land-sea mask.

Included in the Supplementary Information of this paper are palaeorotations such that the modern location of gridcells in the Eocene palaeogeography can be identified, as can the Eocene location of modern gridcells.

We encourage sensitivity studies to the palaeogeography - see Section 4.3.2.

### 4.2.2   Land surface

*(i) vegetation:*

The vegetation in the DeepMIP palaeoclimate simulations should be prescribed as that in H14, which is included digitally as

**Table 2.** Location and filenames of the DeepMIP boundary conditions.

| Simulation Name(s) | Boundary Condition | Location | Filename | Variable Name |
| --- | --- | --- | --- | --- |
| *deepmip-stand-X×CO2*[1] | Topography | Supp Info of H14 | herold_etal_eocene_topo_1x1.nc | topo |
| *deepmip-stand-X×CO2* | Vegetation | Supp Info of H14 | herold_etal_eocene_biome_1x1.nc | eocene_biome[3] |
| *deepmip-stand-X×CO2* | Runoff | Supp Info of H14 | herold_etal_eocene_runoff_1x1.nc | RTM_FLOW_DIRECTI |
| *deepmip-sens-geoggetech* | Topography | Supp Info of Lunt et al. (2016) | bath_ypr.nc, orog_ypr.nc | bathuk, oroguk |
| *deepmip-sens-geogpalmag* | Topography | Supp Info of this paper | Herold2014_TPW.nc | Band1 |

[1] Where *X* can be 3,6, or 12.

[3] 27 biomes. For simplified 11 biomes, use variable eocene_biome-hp.

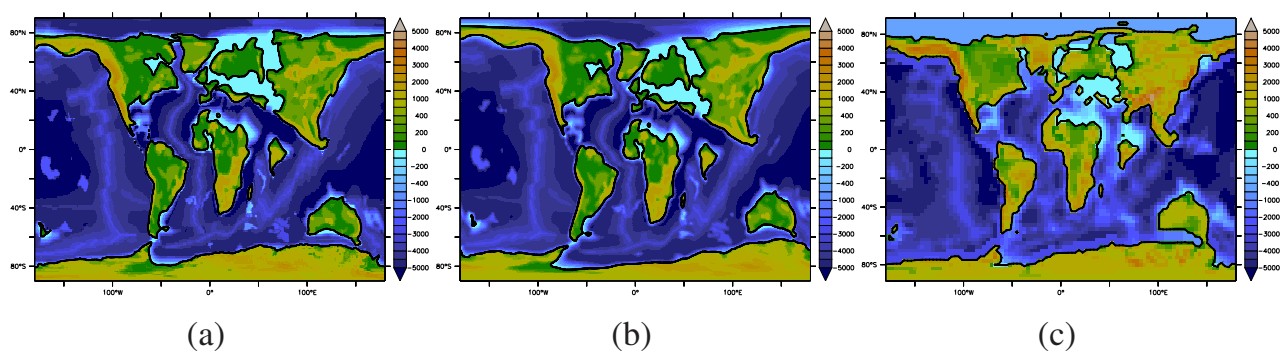

(a)          (b)          (c)

**Figure 3.** Orography and bathymetry for the palaeoclimate simulations in DeepMIP [metres]. (a) The Herold et al. (2014) palaeogeography, as used in the standard palaeoclimate simulations (*deepmip-stand-3×CO2,deepmip-stand-6×CO2,deepmip-stand-9×CO2*). (b) The Herold et al. (2014) palaeogeography, but in the rotation framework given by Torsvik (2011), which is based on a palaeomagnetic reference frame (Baatsen et al., 2016). (c) The Ypresian palaeogeography from Lunt et al. (2016). The location of digital versions of these three palaeo-geographies is given in Table 2.

a netcdf file in the Supplementary Information of H14 (Table 2; note that the BIOME4 vegetation should be used rather than the Sewall vegetation, and that groups may choose to base their vegetation either on the 27 biomes or the 10 megabiomes), and shown here in Fig. 4. Groups should make a lookup table for converting the H14 Eocene dataset to a format that is appropriate for their model. To aid in this process, a modern vegetation dataset is also provided in the Supplementary Information of H14, using the same Plant Functional types as in the H14 Eocene reconstruction; in addition, the lookup table for the CLM (Oleson et al., 2010) land model is provided as a guide.

*(ii) soils:*

Parameters associated with soils should be given constant values over the globe, with values for these parameters (e.g. albedo, water-holding capacity etc.) given by the global-mean of the group's pre-industrial simulation.

*(iii) lakes:*

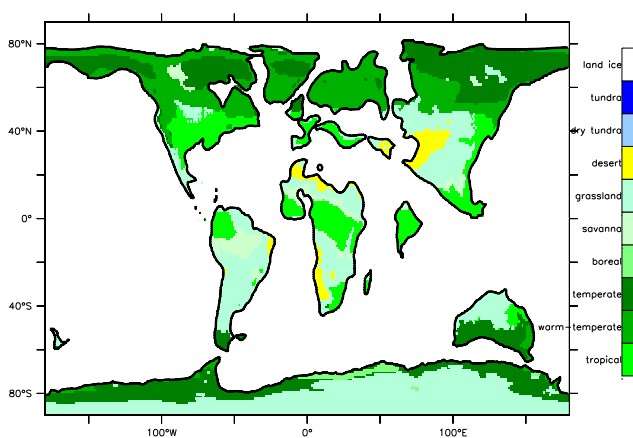

**Figure 4.** Vegetation, expressed as megabiomes, for the palaeoclimate simulations in DeepMIP. A netcdf file of the data at a $1° \times 1°$ resolution is available in the Supplementary Information of Herold et al. (2014) (see Table 2).

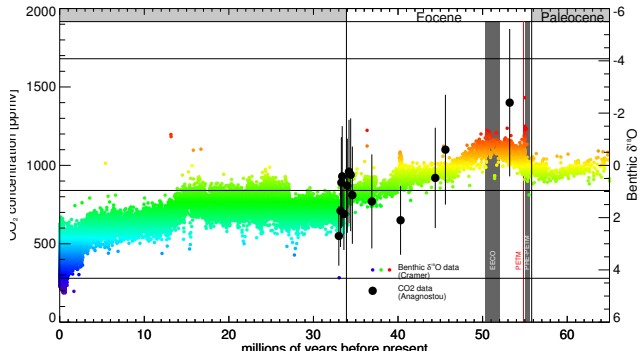

**Figure 5.** Atmospheric $CO_2$ as derived from boron isotopes in the Eocene by (Anagnostou et al., 2016) (black circles and error estimates). Horizontal lines show 280 ppmv (typical preindustrial value), and 840ppmv and 1680 ppmv, corresponding to the *deepmip-stand-3×CO2* and *deepmip-stand-6×CO2* simulations. Also shown are the DeepMIP palaeo intervals - EECO (grey shaded region), pre-PETM (grey shaded region), and the PETM (vertical red line), and the climate evolution over the last 65 million years, as expressed by the benthic oxygen isotope record of Cramer et al. (2009) (coloured dots). Note that the formal definition of the start and end date of each time period is still to be finalised.

No lakes should be prescribed in the DeepMIP palaeoclimate simulations, unless these are predicted dynamically by the model.

*(iv) river runoff:*

River runoff should be taken from the H14 reconstruction, which is included digitally as a netcdf file in the Supplementary Information of H14 (see Table 2).

### 4.2.3  Greenhouse gas concentrations

Each group should carry out three simulations at three different atmospheric $CO_2$ concentrations, expressed as multiples of the value in the pre-industrial simulation (typically $280\,\mathrm{ppmv}$, Section 4.1): (i) $3\times$ pre-industrial (typically $840\,\mathrm{ppmv}$), (ii) $6\times$ pre-industrial (typically $1680\,\mathrm{ppmv}$), and (iii) $12\times$ pre-industrial (typically $3360\,\mathrm{ppmv}$). Assuming a simple relationship between $CO_2$ and temperature, the benthic oxygen isotope record (see Figure 2) implies that, within uncertainty of the $CO_2$ proxies, $CO_2$ concentrations in the EECO and PETM were similar. As such, whereas the low-$CO_2$ simulation can be considered as representing the pre-PETM, the two higher $CO_2$ simulations are intended to represent a range of possible PETM and EECO climate states. The values themselves are based primarily on recent work using boron isotopes (Anagnostou et al., 2016), which indicates that EECO $CO_2$ was $1625\,\mathrm{ppmv}\pm760\,\mathrm{ppmv}$ (Figure 5).

It is thought that non-$CO_2$ greenhouse gases during the early Eocene were elevated relative to pre-industrial, especially $CH_4$ (e.g., $\sim3000\,\mathrm{ppbv}$, Beerling et al., 2011). However, there is considerable uncertainty as to exactly how elevated they were. Given these uncertainties, and the fact that we have chosen to use a modern solar constant as opposed to a reduced solar constant (see Section 4.2.5), which would otherwise offset the $CH_4$ increase, all non-$CO_2$ greenhouse gases and trace gases should be set at the CMIP6 pre-industrial concentrations. In effect, we assume that the $CO_2$ forcing represents the $CO_2$, $CH_4$ (and other non-$CO_2$ greenhouse gases), and solar forcings. For reference, the radiative forcing associated with an increase in $CH_4$ concentrations from preindustrial values to $3000\,\mathrm{ppbv}$ is $+0.98\,\mathrm{Wm}^{-2}$ (Byrne and Goldblatt, 2014), and the radiative forcing associated with an decrease in solar constant from $1361\,\mathrm{Wm}^{-2}$ to $1355.15\,\mathrm{Wm}^{-2}$ (see Sections 4.2.5 and 4.3.5) is $-1.03\,\mathrm{Wm}^{-2}$ (assuming a planetary albedo of 0.3).

Some groups may find the higher $CO_2$ simulations problematic as some models are known to develop a runaway greenhouse at high $CO_2$ (Malte Heinemann, pers comm). In this case, in addition to the $3\times$ simulation, groups can carry out simulations at $2\times$ and $4\times$. In this way, the modelled Eocene climate sensitivity and its nonlinearities can still be investigated.

If groups only have the computational resources to carry out two simulations, they should carry out the $3\times$ and $6\times$ simulations. For groups that can only carry out a single simulation, the analysis of the runs will be limited due to the focus on anomalies in DeepMIP, but we still encourage such groups to participate; in this case they should just carry out the $3\times$ simulation.

For groups with extensive computational resources, we encourage them to carry out additional sensitivity simulations over a range of $CO_2$ values, and in particular at $1\times$, see Section 4.3.1.

### 4.2.4  Aerosols

The representation of aerosols (including mineral dust) in Earth system models is undergoing a period of rapid development. Therefore, we leave the implementation of aerosol fields or emissions rather flexible, and give several options. Groups may choose to (i) leave aerosol distributions or emissions identical to pre-industrial (taking account of the changed land-sea mask), or (ii) treat aerosols prognostically, or (iii) use aerosol concentrations (including mineral dust) from H14, or (iv) use aerosol

30  optical depths from H14, or (v) some combination of the above, depending on the aerosol type. The crucial thing is that groups are asked to document exactly how they have implemented aerosols.

### 4.2.5   Solar constant and astronomical parameters

All simulations should be carried out with the same solar constant and astronomical parameters as in the preindustrial simulation. The solar constant in the CMIP6 *piControl* simulation is defined as $1361.0 \, \mathrm{W \, m^{-2}}$ (Matthes et al., in review, 2016). Although the early Eocene (51 Ma) solar constant was $\sim 0.43\%$ less than this (Gough, 1981), i.e. $\sim 1355 \, \mathrm{W \, m^{-2}}$, we choose to use a modern value in order to (i) aid comparison of any $1 \times CO_2$ simulations (see Section 4.3.1) with pre-industrial, and (b) to offset the absence of elevated $CH_4$ in the experimental design (see Section 4.2.3). As with all of Earth history, astronomical

conditions varied throughout the early Eocene. There is some evidence that the PETM and other Paleogene hyperthermals may have been paced by astronomical forcing (Lourens et al., 2005; Lunt et al., 2011), but the phase of the response relative to the forcing is unknown. The modern orbit has relatively low eccentricity, and so represents a forcing close to the long-term average, and also facilitates comparison with the control pre-industrial simulation. However, we do encourage sensitivity studies to astronomical configuration (see Section 4.3.3).

### 4.2.6   Initial conditions

*(i) Atmosphere and land surface:*
Simulations may be initialised with any state of the atmosphere and land surface, as long as the initial condition would not typically take longer than $\sim 50$ years to spin up in a model with fixed sea surface temperatures; for example, initial snow cover should not be hundreds of metres depth.

*(ii) Ocean:*
Given that even with relatively long simulations, some vestiges of the initial ocean temperature and salinity structure will remain at the end of the simulations, we recommend that all groups adopt the same initialisation procedure for the ocean, but encourage groups to carry out sensitivity studies to the initialisation (see Section 4.3.7). The ocean should be initialised as stationary, with no initial sea ice, and a zonally symmetric temperature ($T$, $^\circ$C) and globally constant salinity ($S$, psu)

distribution given by:

$$T \, [^\circ \mathrm{C}] = \begin{cases} \left( \frac{5000 - z}{5000} \, 25 \cos(\phi) \right) + 15 & \text{if } z \leq 5000 \, \mathrm{m} \\ 15 & \text{if } z > 5000 \, \mathrm{m} \end{cases}$$

$$S \, [\mathrm{psu}] = 34.7 \tag{1}$$

Where $\phi$ is latitude, and $z$ is depth of the ocean (metres below surface).

Some groups have previously found that initialising the model with relatively cold ($< 10 \, ^\circ$C) ocean temperatures at depth

results in a relatively long spinup ($> 5000$ years), due to the suppression of convection – hence the relatively warm initial temperatures at depth prescribed here. Groups for which the recommended initial temperature structure still results in a stratified

ocean with little convection, and hence likely long equilibration timescales (for example those with a model with a particularly high climate sensitivity), may wish to initialise their model with warmer deep ocean temperatures. If so, this should be clearly documented.

The value of $34.7\,\mathrm{psu}$ is the same as the modern mean ocean value. Although the lack of ice sheets in the Eocene would result in a decrease in mean ocean salinity relative to the modern of about $0.6\,\mathrm{psu}$, on these timescales long-term geological sources and sinks of $NaCl$ associated with crustal recycling also play an important role; Hay et al. (2006) estimate mean ocean salinity to be between 35.1 and 36.5 during the Eocene. Given the uncertainties we choose a modern value for simplicity. If groups prefer to initialise salinity with a non-homogeneous distribution, or with a different absolute value, they may do this,

but it should be documented.

For simulations in which oxygen, carbon or other isotopic systems or passive tracers are included, these can be initialised as each individual group sees fit.

### 4.2.7    Length of simulation

Simulations should be carried out for as long as possible. Ideally, simulations should be (a) at least 1000 years in length, and

(b) have an inbalance in the top-of-atmosphere net radiation of less than $0.3\,\mathrm{W\,m^{-2}}$ (or have a similar inbalance to that of the pre-industrial control), and (c) have sea surface temperatures that are not strongly trending (less than $0.1\,^\circ\mathrm{C}$ per century in the global mean). Climatologies should be calculated based on the final 100 years of the simulation.

### 4.2.8    Output format

We strongly recommend that DeepMIP model output should be uploaded to the anticipated PMIP4 component of the CMIP6

database (Eyring et al., 2016), distributed through the Earth System Grid Federation (ESGF). However, if this is not possible, then netcdf files of the variables in Appendix A, including Tables 3-5, should be uploaded to the DeepMIP Model Database, which will be set up if and when required. In any case, for the 'highest priority' variables in Appendix A, Tables 3-5, all months of the simulations should be retained, such that averages can be calculated from arbitrary years of the simulation, and such that equilibrium states can be estimated using the approach of Gregory et al. (2004).

## 4.3    Sensitivity Studies

Sections 4.1 and 4.2 give a summary of the five main simulations. Here we outline some optional sensitivity studies that groups may wish to carry out, although there is no guarantee that other groups will do the same simulations.

### 4.3.1    Sensitivity to $CO_2$

Groups may wish to explore more fully the sensitivity of their model to $CO_2$, and associated non-linearities (e.g. Caballero

and Huber, 2013), by carrying out additional simulations over a range of $CO_2$. Normally these would be multiples of the pre-industrial concentration, in addition to the standard $3\times$, $6\times$, and $12\times$ simulations. In particular, we encourage groups to

carry out a $1\times$ simulation, for comparison with the pre-industrial control – this simulation enables the contribution of non-$CO_2$ forcings (palaeogeography and ice sheets) to early Eocene warmth to be evaluated.

### 4.3.2 Sensitivity to palaeogeography

Getech Group plc (www.getech.com) have provided an alternative palaeogeographic reconstruction that may be used for sensitivity studies, in particular simulation *deepmip-sens-geoggetech* (see Tables 1,2). It is included digitally in Lunt et al. (2016) as a netcdf file at a resolution of $3.75°$ longitude $\times 2.5°$ latitude, and is shown in Figure 3(c). Because a high resolution version of this topography is not available, groups will need to use the subgridscale palaeogeography from the H14 reconstruction, and interpolate to the new land-sea mask as appropriate. The vegetation, river routing etc. from H14 will also need to be extrapolated to the new land-sea mask. Ideally, groups would carry out these simulations at the same three $CO_2$ levels as in the standard simulations, but if groups can only carry out a limited number of simulations with this palaeogeography, they should carry them out in the following order of priority (highest priority first): $3\times, 6\times, 12\times$.

Both Getech and H14 use the plate rotation model of Müller et al. (2008), which is derived from relative plate motions tied to a mantle reference frame. van Hinsbergen et al. (2015) argue that for palaeoclimate studies, plate motions should be tied to the spin axis of the Earth using a palaeomagnetic reference frame in order to obtain accurate estimates of palaeolatitude. For this reason, we also provide an additional version of the H14 palaeogeography, but rotated to a palaeomagnetic reference frame based on the methods outlined by van Hinsbergen et al. (2015) and Baatsen et al. (2016), for use in sensitivity study *deepmip-sens-geogpalmag* (see Tables 1,2). This is shown in Figure 3(b), and provided in the Supplementary Information to this paper.

Furthermore, some of the topographic features could have evolved significantly throughout the $\sim$55-51 Ma period of interest, making it unlikely that a single palaeogeography can represent all the DeepMIP time periods to the same extent. Groups are therefore encouraged to carry out sensitivity studies around the H14 palaeogeography, to explore the uncertainties in climate which may result from uncertainties in the spatial and temporal evolution of different topographic features. These studies may include widening/constricting and shallowing/deepening key ocean gateways, changing the bathymetry and extent of ocean shelves, and raising/lowering mountain ranges. In particular, we encourage groups to carry out sensitivity studies in which the NE Atlantic-Arctic gateway to the east of Greenland is closed. This is because there is evidence that a short, transient period of $\sim$kilometer-scale tectonic uplift of NW Europe and Greenland, associated with the North Atlantic Large Igneous Province, severely restricted the NE Atlantic-Arctic oceanic gateway during the PETM period in comparison with the pre-PETM and EECO periods (Hartley et al., 2011; Jones and White, 2003; Maclennan and Jones, 2006; Saunders et al., 2007).

### 4.3.3 Sensitivity to astronomical parameters

Evidence of cyclicity during the Paleocene and early Eocene indicates that a component of the warmth of the PETM may be astronomically forced (Lourens et al., 2005; Westerhold et al., 2007; Galeotti et al., 2010). As such, we encourage sensitivity studies to astronomical configuration. As the standard DeepMIP palaeoclimate simulations are configured with a modern orbit,

which has relatively low eccentricity, we suggest groups carry out additional simulations with high eccentricity ($e = 0.054$ compared with a modern value of $e = 0.017$), and northern hemisphere winter corresponding with both aphelion and perihelion.

### 4.3.4 Sensitivity to vegetation

Those groups which have a model that includes dynamic vegetation may carry out sensitivity studies with dynamic vegetation turned on. The initial condition should be broadleaf or needleleaf trees at all locations. Ideally groups would carry out these simulations at the same three $CO_2$ levels as in the standard simulations, but if groups can only carry out a limited number of simulations with the dynamic vegetation, they should carry them out in the following order of priority (highest priority first): $3\times, 6\times, 12\times$. Groups with models that include a dynamic vegetation component can choose to pass to their vegetation model either the ambient atmospheric $CO_2$, or a lower concentration if required for model stability.

### 4.3.5 Sensitivity to solar constant

Groups may wish to explore the relative radiative forcing of the solar luminosity compared with other forcings, by carrying out an Eocene simulation with a reduced solar luminosity. The suggested reduction is 0.43% (Gough, 1981), which would normally be from $1361.0 \, \mathrm{W \, m^{-2}}$ in the modern to $1355.15 \, \mathrm{W \, m^{-2}}$ in the Eocene. This would typically be carried out at a $CO_2$ level of $3\times$.

### 4.3.6 Sensitivity to non-$CO_2$ greenhouse gases

Groups may choose to explore sensitivity to non-$CO_2$ greenhouse gases (see Section 4.2.3 for discussion of $CH_4$), in particular if these can be predicted by the model interactively.

### 4.3.7 Sensitivity to initialisation

We encourage groups to carry out sensitivity studies to the initialisation of the ocean temperature and salinity. It is possible that models will exhibit bistability with respect to initial condition, and as discussed in Section 4.2.6 we expect that the equilibration time will be a function of the initial conditions and will be different for different models.

### 4.3.8 'Best in Show'

Participants are invited to carry out simulations in which they attempt to best match existing climate proxy data. This may be done in a number of ways, for example by modifying the aerosols (Huber and Caballero, 2011), cloud properties (Kiehl and Shields, 2013), physics parameters (Sagoo et al., 2013), using very high $CO_2$ (Huber and Caballero, 2011), incorporating dynamic vegetation (Loptson et al., 2014), modifying gateways (Roberts et al., 2009), modifying orbital configuration, including non-$CO_2$ greenhouse gases, or a combination of the above and other modifications.

## 5 Climate Proxies

A major focus of DeepMIP will be to develop a new synthesis of climate proxy data for the latest Paleocene and early Eocene, focussing on the three targetted time intervals: pre-PETM, PETM and EECO. The main focus of DeepMIP will be on temperature and precipitation proxies. Two working groups have been set up to compile these data from marine and terrestrial records. These groups will also work together to generate new data sets for poorly documented regions, such as the tropics, and will seek multiple lines of evidence for climate reconstructions wherever possible. The marine working group is excited by the possibility of using innovative analytical techniques (e.g. Kozdon et al., 2013) to recover robust estimates for sea surface temperature from planktic foraminiferal assemblages within legacy sediment cores of the International Ocean Discovery Program. Published data sets will be combined into an open-access online database. The EECO and PETM/pre-PETM marine compilations of Lunt et al. (2012), Hollis et al. (2012), and Dunkley Jones et al. (2013), and EECO terrestrial compilations of Huber and Caballero (2011) provide a starting point for this database. One of the great challenges for these working groups will be to develop new ways to assess climate proxy reliability and quantify uncertainties. In some cases, it may be more straightforward to consider relative changes in proxies rather than report absolute values. Climate proxy system modelling (Evans et al., 2013) coupled with Bayesian analysis (e.g. Khider et al., 2015; Tierney and Tingley, 2014) has great potential for improving estimation of uncertainties and directly linking our climate proxy compilation with the climate simulations. In addition to these quantitative estimates of uncertainty, all data will be qualitatively assessed based on expert opinion, for example characterising proxies as high, medium, or low confidence (as has been done in PlioMIP, see Dowsett et al., 2012).

We anticipate a companion paper to this one in which we will give more details of the DeepMIP data and associated protocols.

## 6 Products

In addition to this experimental design paper, and papers describing the new climate proxy syntheses, once the model simulations are complete we anticipate producing overarching papers describing the 'large-scale features' of the model simulations, and model–data comparisons. Following this, we anticipate a number of spin-off papers looking at various other aspects of the model simulations (e.g., ENSO, ocean circulation, monsoons). In particular we expect papers that explore the relevance of the DeepMIP simulations and climate proxy syntheses for future climate, for example through model developments that arise as a result of the model-data comparison, or emergent constraints (Bracegirdle and Stephenson, 2013) on global-scale metrics such as climate sensitivity. Furthermore, we will encourage modelling participants to publish individual papers that describe their own simulations in detail, including how the boundary conditions were implemented. In this respect, we are basing our dissemination strategy on that of PlioMIP (Haywood et al., 2013); see their Special Issue at http://www.geosci-model-dev.net/special_issue5.html.

## 7 Data availability

The boundary conditions for the standard DeepMIP palaeoclimate simulations are supplied as Supplementary Information in H14 (Herold et al., 2014); see Table 2. For availability of boundary conditions for DeepMIP sensitivity studies, also see Table 2. Data held in both the CMIP6 and DeepMIP Model databases, when these are operational, will likely be freely accessible through data portals after registration.

## Appendix A: Output variables

As stated in Section 4.2.8, we strongly reccommend that model output is uploaded to the CMIP6 database. If the CMIP6 database cannot be used, the variables in Tables 3-5 should be submitted to the DeepMIP Model Database, which will be set up if and when required. Climatological averages of the final 100 years of the simulation should be supplied for each month (12 fields for each variable). In addition, for the highest priority variables, all months of the simulation should be supplied.

Furthermore, as many groups are interested in hydrological extremes, groups should aim to produce ten years of hourly precipitation, evaporation and runoff data.

*Author contributions.* A first draft of this paper was written by Dan Lunt and Matt Huber. It was subsequently edited based on discussions at a DeepMIP meeting in January 2016 at NCAR, Boulder, Colorado, USA, and following further email discussions with the DeepMIP community. All authors contributed at the meeting and/or in the subsequent email discussions.

*Acknowledgements.* We thank NERC grant NE/N006828/1 for providing funds for the first DeepMIP meeting in Boulder, Colorado, USA, in January 2016. DJL acknowledges NERC grant "Cretaceous-Paleocene-Eocene: Exploring Climate and Climate Sensitivity" (NE/K014757/1), and advanced ERC grant "The Greenhouse Earth System" (T-GRES, project reference 340923), awarded to Rich Pancost. We thank two anonymous reviewers whose comments were very useful in improving and clarifying the experimental design.

**Table 3.** Atmosphere variables

| Variable | Units | Highest priority |
|---|---|---|
| Near surface (1.5 m) air temperature | °C | X |
| Surface skin temperature | °C | |
| Precipitation | $\mathrm{kg\,m^2\,s^{-1}}$ | X |
| Total evaporation | $\mathrm{kg\,m^2\,s^{-1}}$ | |
| Total cloud cover | [0,1] | |
| FLNS | $\mathrm{W\,m^{-2}}$ | |
| FLNT | $\mathrm{W\,m^{-2}}$ | X |
| FSDS | $\mathrm{W\,m^{-2}}$ | |
| FSNS | $\mathrm{W\,m^{-2}}$ | |
| FSNT | $\mathrm{W\,m^{-2}}$ | X |
| FSDT | $\mathrm{W\,m^{-2}}$ | |
| sensible heat flux | $\mathrm{W\,m^{-2}}$ | |
| latent heat flux | $\mathrm{W\,m^{-2}}$ | |
| Near surface (10 m) u wind | $\mathrm{m\,s^{-1}}$ | |
| Near surface (10 m) v wind | $\mathrm{m\,s^{-1}}$ | |
| surface wind stress (x) | $\mathrm{N\,m^{-2}}$ | |
| surface wind stress (y) | $\mathrm{N\,m^{-2}}$ | |
| mean sea-level pressure | Pa | |
| surface pressure | Pa | |
| u winds on model atmospheric levels | $\mathrm{m\,s^{-1}}$ | |
| v winds on model atmospheric levels | $\mathrm{m\,s^{-1}}$ | |
| w winds on model atmospheric levels | $\mathrm{m\,s^{-1}}$ | |
| u wind at 200 mbar | $\mathrm{m\,s^{-1}}$ | |
| v wind at 200 mbar | $\mathrm{m\,s^{-1}}$ | |
| u wind at 500 mbar | $\mathrm{m\,s^{-1}}$ | |
| v wind at 500 mbar | $\mathrm{m\,s^{-1}}$ | |
| u wind at 850 mbar | $\mathrm{m\,s^{-1}}$ | |
| v wind at 850 mbar | $\mathrm{m\,s^{-1}}$ | |
| geopotential height at 200 mbar | m | |
| geopotential height at 500 mbar | m | |
| geopotential height at 850 mbar | m | |
| temperature at 200 mbar | °C | |
| temperature at 500 mbar | °C | |
| temperature at 850 mbar | °C | |
| specific humidity at 200 mbar | $\mathrm{kg\,kg^{-1}}$ | |
| specific humidity at 500 mbar | $\mathrm{kg\,kg^{-1}}$ | |
| specific humidity at 850 mbar | $\mathrm{kg\,kg^{-1}}$ | |

N.B. F$XYZ$ notation

F = flux

$X$ = S(hortwave) or L(ongwave)

$Y$ = D(own) or N(et)

$Z$ = S(urface) or T(op of atmosphere)

# References

Anagnostou, E., John, E., Edgar, K., Foster, G., Ridgwell, A., Inglis, G., Pancost, R., Lunt, D., and Pearson, P.: Changing atmospheric CO2 concentration was the primary driver of early Cenozoic climate, Nature, doi:10.1038/nature17423, 2016.

**Table 4.** Ocean Variables

| Variable | Units | Highest priority |
|---|---|---|
| sea surface temperature | $^\circ$C | X |
| sea-ice fraction | [0,1] | X |
| u,v,w on model levels | $\mathrm{cm\,s^{-1}}$ | |
| potential temperature on model levels | $^\circ$C | |
| salinity on model levels | psu | |
| barotropic streamfunction | $\mathrm{cm^3\,s^{-1}}$ | |
| mixed-layer depth | m | |
| global overturning streamfunction | Sv | |

**Table 5.** Boundary conditions

| Variable | Units |
|---|---|
| land-sea mask | [0,1] |
| topography | m |
| bathymetry | m |

Baatsen, M., van Hinsbergen, D. J. J., von der Heydt, A. S., Dijkstra, H. A., Sluijs, A., Abels, H. A., and Bijl, P. K.: Reconstructing
geographical boundary conditions for palaeoclimate modelling during the Cenozoic, Climate of the Past, 12, 1635–1644, doi:10.5194/cp-12-1635-2016, http://www.clim-past.net/12/1635/2016/, 2016.

Beerling, D. J., Fox, A., Stevenson, D. S., and Valdes, P. J.: Enhanced chemistry-climate feedbacks in past greenhouse worlds, Proceedings of the national Academy of Sciences, 108, 9770–9775, 2011.

Bracegirdle, T. J. and Stephenson, D. B.: On the Robustness of Emergent Constraints Used in Multimodel Climate Change Projections of
Arctic Warming, Journal of Climate, 26, 669–678, 2013.

Byrne, B. and Goldblatt, C.: Radiative forcing at high concentrations of well-mixed greenhouse gases, Geophysical Research Letters, 41, 152–160, 2014.

Caballero, R. and Huber, M.: State-dependent climate sensitivity in past warm climates and its implications for future climate projections, PNAS, 110, 14 162–14 167, 2013.

Carmichael, M. J., Lunt, D. J., Huber, M., Heinemann, M., Kiehl, J., LeGrande, A., Loptson, C. A., Roberts, C. D., Sagoo, N., Shields, C., Valdes, P. J., Winguth, A., Winguth, C., and Pancost, R. D.: A model–model and data–model comparison for the early Eocene hydrological cycle, Climate of the Past, 12, 455–481, doi:10.5194/cp-12-455-2016, http://www.clim-past.net/12/455/2016/, 2016.

Cramer, B. S., Toggweiler, J. R., Wright, J. D., Katz, M. E., and Miller, K. G.: Ocean overturning since the Late Cretaceous: Inferences from a new benthic foraminiferal isotope compilation, Paleoceanography, 24, doi:10.1029/2008PA001 683, 2009.

Dowsett, H. J., Robinson, M. M., Haywood, A. M., Hill, D. J., Dolan, A. M., Stoll, D. K., Chan, W.-L., Abe-Ouchi, A., Chandler, M. A., Rosenbloom, N. A., Otto-Bliesner, B. L., Bragg, F. J., Lunt, D. J., Foley, K. M., and Riesselman, C. R.: Assessing confidence in Pliocene sea surface temperatures to evaluate predictive models, Nature Climate Change, 2, 365–371, 2012.

Dunkley Jones, T., Lunt, D., Schmidt, D., Ridgwell, A., Sluijs, A., Valdes, P., and Maslin, M.: Climate model and proxy data constraints on ocean warming across the Paleocene-Eocene Thermal Maximum, Earth Science Reviews, p. doi.org/10.1016/j.earscirev.2013.07.004,
2013.

Evans, M. N., Tolwinski-Ward, S. E., Thompson, D. M., and Anchukaitis, K. J.: Applications of proxy system modeling in high resolution paleoclimatology, Quaternary Science Reviews, 76, 16–28, 2013.

Eyring, V., Bony, S., Meehl, G. A., Senior, C. A., Stevens, B., Stouffer, R. J., and Taylor, K. E.: Overview of the Coupled Model Intercomparison Project Phase 6 (CMIP6) experimental design and organization, Geoscientific Model Development, 9, 1937–1958, doi:10.5194/gmd-
9-1937-2016, http://www.geosci-model-dev.net/9/1937/2016/, 2016.

Galeotti, S., Krishnan, S., Pagani, M., Lanci, L., Gaudio, A., Zachos, J. C., nd G. Morelli, S. M., and Lourens, L.: Orbital chronology of Early Eocene hyperthermals from the Contessa Road section, central Italy, Earth and Planetary Science Letters, 290, 192–200, 2010.

Gasson, E., Lunt, D. J., DeConto, R., Goldner, A., Heinemann, M., Huber, M., LeGrande, A. N., Pollard, D., Sagoo, N., Siddall, M., Winguth, A., and Valdes, P. J.: Uncertainties in the modelled $CO_2$ threshold for Antarctic glaciation, Climate of the Past, 10, 451–466,
doi:10.5194/cp-10-451-2014, http://www.clim-past.net/10/451/2014/, 2014.

Gent, P., Danabasoglu, G., Donner, L., Holland, M., Hunke, E., Jayne, S., Lawrence, D., Neale, R., Rasch, P., Vertenstein, M., Worley, P., Yang, Z.-L., and Zhang, M.: The Community Climate System Model Version 4, J. Climate, 24, 49734 991, doi:10.1175/2011jcli4083, 2011.

Gough, D.: Solar interior structure and luminosity variations, Sol, Phys., 74, 21–34, 1981.

Gregory, J. M., W. J. Ingram, W. J., Palmer, M. A., Jones, G. S., Stott, P. A., Thorpe, R. B., Lowe, J. A., Johns, T. C., and Williams, K. D.: A new method for diagnosing radiative forcing and climate sensitivit, Geophysical Research Letters, 31, L03 205, doi:10.1029/2003gl018747, 2004.

Hansen, J., Sato, M., Russell, G., and Kharecha, P.: Climate sensitivity, sea level and atmospheric carbon dioxide, Phil. Trans. R. Soc. A, 371, 2013.

Hartley, R. A., Roberts, G., White, N., and Richardson, C. J.: Transient convective uplift of an ancient buried landscape, Nature Geoscience, 4, 562–565, 2011.

Hay, W. W., Migdisov, A., Balukhovsky, A. N., Wold, C. N., Flögel, S., and Söding, A.: Evaporites and the salinity of the ocean during the Phanerozoic: Implications for climate, ocean circulation and life, Palaeogeography, Palaeoclimatology, Palaeoecology, 240, 3–46, 2006.

Haywood, A. M., Hill, D. J., Dolan, A. M., Otto-Bliesner, B. L., Bragg, F., Chan, W.-L., Chandler, M. A., Contoux, C., Dowsett, H. J.,
Jost, A., Kamae, Y., Lohmann, G., Lunt, D. J., Abe-Ouchi, A., Pickering, S. J., Ramstein, G., Rosenbloom, N. A., Salzmann, U., Sohl, L., Stepanek, C., Ueda, H., Yan, Q., and Zhang, Z.: Large-scale features of Pliocene climate: results from the Pliocene Model Intercomparison Project, Climate of the Past, 9, 191–209, doi:10.5194/cp-9-191-2013, http://www.clim-past.net/9/191/2013/, 2013.

Herold, N., Buzan, J., Seton, M., Goldner, A., Green, J. A. M., Müller, R. D., Markwick, P., and Huber, M.: A suite of early Eocene ( 55 Ma) climate model boundary conditions, Geoscientific Model Development, 7, 2077–2090, doi:10.5194/gmd-7-2077-2014, http:
30   //www.geosci-model-dev.net/7/2077/2014/, 2014.

Hollis, C. J., Taylor, K. W. T., Handley, L., Pancost, R. D., Huber, M., Creech, J., Hines, B., Crouch, E. M., Morgans, H. E. G., Crampton, J. S., Gibbs, S., Pearson, P., and Zachos, J. C.: Early Paleogene temperature history of the Southwest Pacific Ocean: reconciling proxies and models, Earth and Planetary Science Letters, 349-350, 53–66, 2012.

Huber, M. and Caballero, R.: The early Eocene equable climate problem revisited, Climate of the Past, 7, 603–633, 2011.

Jones, S. M. and White, N. J.: Shape and size of the initiating Iceland Plume swell, Earth and Planetary Science Letters, 216, 271–282, 2003.

Khider, D., Huerta, G., Jackson, C., Stott, L. D., and Emile-Geay, J.: A Bayesian, multivariate calibration for Globigerinoides ruber Mg/Ca, Geochem. Geophys. Geosyst., 16, 2916–2932, doi:10.1002/2015GC005844, 2015.

Kiehl, J. T. and Shields, C. A.: Sensitivity of the PalaeoceneEocene Thermal Maximum climate to cloud properties, Phil. Trans. R. Soc. A, 371, doi:10.1098/rsta.2013.0093, 2013.

Kozdon, R., Kelly, D. C., Kitajima, K., Strickland, A., Fournelle, J. H., and Valley, J. W.: In situ d18O and Mg/Ca analyses of diagenetic and planktic foraminiferal calcite preserved in a deep-sea record of the Paleocene-Eocene Thermal Maximum, Paleoceanography, 28, 517–528, doi:10.1002/palo.20048, 2013.

Loptson, C. A., Lunt, D. J., and Francis, J. E.: Investigating vegetation-climate feedbacks during the early Eocene, Clim. Past, 10, 419–436, 2014.

Lourens, L. J., Sluijs, A., Kroon, D., Zachos, J. C., Thomas, E., Rohl, U., Bowles, J., and Raffi, I.: Astronomical pacing of late Palaeocene to early Eocene global warming events, Nature, 435, 1083–1087, 2005.

Lunt, D., Ridgwell, A., Sluijs, A., Zachos, J., Hunter, S., and Haywood, A.: A model for orbital pacing of methane hydrate destabilization during the Palaeogene, Nature Geoscience, 4, 775–778, 2011.

Lunt, D. J., Dunkley Jones, T., Heinemann, M., Huber, M., LeGrande, A., Winguth, A., Loptson, C., Marotzke, J., Roberts, C. D., Tindall, J., Valdes, P., and Winguth, C.: A model-data comparison for a multi-model ensemble of early Eocene atmosphere-ocean simulations: EoMIP, Climate of the Past, 8, 1717–1736, 2012.

Lunt, D. J., Elderfield, H., Pancost, R., Ridgwell, A., Foster, G., Haywood, A., Kiehl, J., Sagoo, N., and Stone, E.J. Valdes, P.: Warm climates of the past - a lesson for the future?, Phil. Trans. R. Soc. A, 371, doi:10.1098/rsta.2013.0146, 2013.

Lunt, D. J., Farnsworth, A., Loptson, C., Foster, G. L., Markwick, P., O'Brien, C. L., Pancost, R. D., Robinson, S. A., and Wrobel, N.: Palaeogeographic controls on climate and proxy interpretation, Climate of the Past, 12, 1181–1198, doi:10.5194/cp-12-1181-2016, http://www.clim-past.net/12/1181/2016/, 2016.

Maclennan, J. and Jones, S. M.: Regional uplift, gas hydrate dissociation and the origins of the Paleocene-Eocene Thermal Maximum, Earth and Planetary Science Letters, 245, 65–80, 2006.

Matthes, K., Funke, B., Anderson, M. E., Barnard, L., Beer, J., Charbonneau, P., Clilverd, M. A., Dudok de Wit, T., Haberreiter, M., Hendry, A., Jackman, C. H., Kretschmar, M., Kruschke, T., Kunze, M., Langematz, U., Marsh, D. R., Maycock, A., Misios, S., Rodger, C. J., Scaife, A. A., Seppälä, A., Shangguan, M., Sinnhuber, M., Tourpali, K., Usoskin, I., van de Kamp, M., Verronen, P. T., and Versick, S.: Solar Forcing for CMIP6 (v3.1), Geoscientific Model Development Discussions, 2016, 1–82, doi:10.5194/gmd-2016-91, http://www.geosci-model-dev-discuss.net/gmd-2016-91/, in review, 2016.

Müller, R. D., Sdrolias, M., Gaina, C., and Roest, W. R.: Age, spreading rates, and spreading asymmetry of the world's ocean crust, Geochem. Geophys. Geosyst., 9, Q04 006, doi:10.1029/2007GC001743, 2008.

Oleson, K., Lawrence, D., Bonan, G., Flanner, M., Kluzek, E., Lawrence, P., Levis, S., Swenson, S., Thornton, P., Dai, A., Decker, M., Dickinson, R., Feddema, J., Heald, C., Hoffman, F., Lamarque, J.-F., Mahowald, N., Niu, G.-Y., Qian, T., Randerson, J., Running, S., Sakaguchi, K., Slater, A., Stockli, R., Wang, A., Yang, Z.-L., Zeng, X., and Zeng, X.: Technical Description of version 4.0 of the Community Land Model (CLM), NCAR Technical Note NCAR/TN-478+STR, p. 257pp, 2010.

Roberts, C. D., LeGrande, A. N., and Tripati, A. K.: Climate sensitivity to Arctic seaway restriction during the early Paleogene, Earth and Planetary Science Letters, 286, 576–585, 2009.

Sagoo, N., Valdes, P., Flecker, R., and Gregoire, L.: The Early Eocene equable climate problem: can perturbations of climate model parameters identify possible solutions?, Phil. Trans. R. Soc. A, 371, doi:10.1098/rsta.2013.0123, 2013.

Saunders, A. D., Jones, S. M., Morgan, L. A., Pierce, K. L., Widdowson, M., and Xu, Y.: The role of mantle plumes in the formation of continental large igneous provinces: Field evidence used to constrain the effects of regional uplift, Chemical Geology, 241, 282–318, 2007.

Tierney, J. E. and Tingley, M. P.: A Bayesian, spatially-varying calibration model for the TEX86 proxy, Geochimica et Cosmochimica Acta, 127, 83–106, doi:83-106, 2014.

van Hinsbergen, D. J. J., de Groot, L. V., van Schaik, S. J., Spakman, W., Bijl, P. K., Sluijs, A., Langereis, C. G., and Brinkhuis, H.: A Paleolatitude Calculator for Paleoclimate Studies, PLoS ONE, 10, e0126 946, doi:10.1371/journal.pone.0126946, 2015.

Westerhold, T., Rohl, U., Laskar, J., Raffi, I., Bowles, J., Lourens, L. J., and Zachos, J. C.: On the duration of magnetochrons C24r and C25n and the timing of early Eocene global warming events: Implications from the Ocean Drilling Program Leg 208 Walvis Ridge depth transect, Paleoceanography, 22, PA2201, 2007.

Zeebe, R. E., Ridgwell, A., and Zachos, J. C.: Anthropogenic carbon release rate unprecedented during the past 66 million years, Nature Geoscience, 9, 325–329, 2016.