# Peer review of "DeepMIP: experimental design for model simulations of the EECO, PETM, and pre-PETM."

_Geoscientific Model Development, 2016_

## Short Comment (SC1) · 4 Jul 2016

Dear Authors,

in case of articles belonging to the CMIP6 model intercomparison, we established that CMIP6 should be named in the title to help the readers to directly associate the article with the correct model intercomparison project.

Therefore I like to encourage you, to follow the form of the CMIP6 article title structure and add "for PMIP" to the title: e.g., "DeepMIP for PMIP: experimental design for model simulations of the EECO, PETM, and pre-PETM"

Best regards, Astrid Kerkweg

---

## Short Comment (SC2) · 4 Jul 2016

Dear Jules,

I know, that it is not a CMIP6 paper, but it is (see abstract) a PMIP paper. Therefore I suggested to adopt the style, clarifying already in the title that this is a PMIP paper.

Best regards, Astrid

---

## Editor Comment (EC1) · J. C. Hargreaves (Editor) · 4 Jul 2016

Hi Astrid - Thanks for your diligence! However, this paper is not in CMIP6 Special Issue (and indeed DeepMIP is not part of CMIP6). jules
* * *

---

## Editor Comment (EC2) · J. C. Hargreaves (Editor) · 4 Jul 2016

Oh yes, good idea! Let's make it "PMIP4", though. It seems likely there will be a number of other PMIP4 papers being submitted to GMD in the next few months, and the one that is already well through the review process (Ivanovic et al, on the deglaciation) does indeed include "PMIP4" in the title. jules

———————————————

---

## Referee Comment (RC1) · Anonymous Referee #1 · 9 Aug 2016

General comments
* * *
This manuscript describes the DeepMIP protocol for common climate simulations for periods older than the Pliocene. These periods, the Early Eocene Climate Optimum (EECO), the Paleocene-Eocene Thermal Maximum (PETM) and the period immediately prior to the PETM, are among the warmest period known in the Earth climate history. This makes them interesting to model in the context of better understanding processes at work in warm climates, such as the future one could be, and of testing the models' ability to represent these warm climates.

The paper is generally clearly written and the justification and experimental design of the experiments are quite convincing as far as these past climates and comparison to

available data is concerned. It is proposed to run sensitivity experiments to account for different sets of boundary conditions, and experiments with different CO2 levels to account for the uncertainties in the CO2 reconstructions, which is interesting to test climate sensitivity to CO2 under these conditions. With this perspective, the experimental design could be improved to better liaise with the CMIP6 exercise. In particular, the CMIP6 DECK includes a preindustrial and an abrupt4xCO2 experiment, in which the CO2 level is quadrupled from the pre-industrial level. The DeepMIP protocol recommends to run the pre-industrial as in CMIP6 but it would be very interesting for the groups to also run the CMIP6 DECK abrupt4xCO2 simulation. If the DeepMIP protocol also included a similar 4xCO2 experiment with the deep-time continents and ocean, then it would be easy to examine whether the deep time continents and oceans have an impact on the Earth's sensitivity to greenhouse gases increase, and how much can be inferred on climate sensitivity from these climates. I would therefore argue for changing the priorities in the experimental design (cf page 7, lines 9-11) and to test 2x and 4xCO2 (and higher) first, rather than 3x and 6xCO2. The pre-industrial control and the abrupt4xCO2 experiments have also been proposed to be mandatory for modelling groups wishing to take part in the PMIP4 exercise, to better liaise with CMIP6, so the above recommendation would also warrant a better relation to PMIP4 activities.

Apart from the more specific comments below, what is missing from the manuscript at this stage is a table summarizing the experiments and boundary conditions, and the names given to the experiments so that all groups use these names. Additional figures could also be inserted to better illustrate the scope of the project and the different options in boundary conditions, as explained in the comments below.

Otherwise, the paper is clear and well written and can be published after these corrections are made.
* * *
Specific comments
* * *
pages 2-3, section 2: this section on previous work could be illustrated by a figure showing what can be improved from this previous work.

page 3, section 3: for outsiders, it would be good to have a figure locating the three periods in a broader chronology of the Earth climate evolution.

page 4, section 4.2, lines 11-12: "There are three standard simulations" seem to contrast with the sentence at the top of the page: "The DeepMIP experimental protocol consists of four main simulations". It would be good to clarify this: three periods, but four simulations.

page 4, section 4.2.1: it would be good to stress at this point that the same paleogeography is used for all three periods. Also, the main cautionary points in the implementation of the paleogeography, such as straits and shallow basins, should be highlighted, in relation with the sensitivity experiments proposed in section4.3.2.

page 5, lines 7ff, about the soils: I am not very familiar with this issue, but I would expect spatial heterogeneities in soil properties, so how can these be prescribed "homogeneously"?

page 7, value of the solar constant: it has been revised to 1361 W/m2 (Matthes et al, http://www.geosci-model-dev-discuss.net/gmd-2016-91/). Since this paper is still in discussion, it will be worth referring to its final value when it is out. However, this has an impact on the discussion about early Eocene values in section 4.2.5 and on the sensitivity experiments proposed in section 4.3.5. Is the value found by Gough (1981) actually tied to a present value of 1365 W/m2?

page 7, justification of not changing the solar constant in the DeepMIP experiment, to counteract the absence of elevated CH4 in the design. This should be better justified. Both forcings are not equivalent and it is rather easy to change the CH4 values in the models. At least the radiative forcing from the CH4 high values should be evaluated

and compared to the non-changes in the solar constant.

page 9: sensitivity to paleogeography: maps of differences could be shown to convince modelling groups that it is worth investing the time to perform these sensitivity experiments. A practical question is about where to actually find this other paleogeography.

page 12: the PMIP data base should be used ! this is the only way cross-period analyses can be performed and other groups can be involved, bringing additional diagnostics and analyses. So the list in Table 1 should be expressed in terms of PMIP/CMIP6 variables. In particular, the acronyms "FLNS", "FLNT" etc should be explained.

——————————

Typos/rephrasing

—————————-

page 2, line 7. Replace "paleo simulations" by "paleoclimate simulations" (we hope that the simulations are new, and not "paleo")

page 2, line 15: "deep-time model intercomparison project". should this be "deep-time climates"? The project does not aim at comparing deep times, but rather their climates, doesn't it?

pages 4 and 5: references should be added for the CESM and CLM models.

page7, line 25: the Louvain-la-Neuve group has recommended to use the term "astronomical parameters" rather than "orbital parameters" since obliquity is not an element describing the orbit of the Earth.

page 9, line 9: reference to Appendix 1 should be changed to Appendix A.

page 10, line 27: the link to the section is missing

page 11, last line: parentheses are missing around the web site reference.

---

## Referee Comment (RC2) · Anonymous Referee #2 · 17 Aug 2016

SUMMARY:

This manuscript describes the 'DeepMIP' (PMIP) protocol for palaeoclimate simulations of the latest Palaeocene and Eocene.

DeepMIP is a valuable addition to the world of MIPs, as eloquently and succinctly summarised by the authors: 'models of past high-$CO_2$ [> 800 ppmv] periods have never been evaluated in a consistent framework'. The manuscript is clearly written, flows well and has a logical structure. The protocol itself has no doubt taken a lot of in depth discussion to finalise, and comes across as having been thoroughly designed. However, there are a few boundary conditions (solar constant and $CH_4$) that need updating. In some places, some expansion of the text is required to clearly explain what may already be apparent to experts immersed in the science, but would be helpful information for

the less well-versed. These mainly relate to summarising existing literature and would not be fundamental changes to the manuscript structure or protocol details. In addition to the model products, the new [syntheses of] geological records will be an important result.

In summary, this is a very well written manuscript that is enjoyable to read and presents a robust and much needed protocol for simulating the latest Palaeocene-Eocene climate. With minor changes, I believe it is well suited for publication in GMD, and I look forward to the MIP results.

General and specific comments follow.

GENERAL COMMENTS:

1. There needs to be better consistency between the way the core simulations are referred to: a. Whether there are 3 or 4 (I understand that there are 3 palaeo simulations and 1 preindustrial simulation and that these are the core, but this is not clear enough in the manuscript when interchanging between describing 3 and 4 core simulations): b. How the palaeoclimate simulations are named as both 'pre-PETM', 'PETM' and 'EECO' versus 'two early Eocene, and one latest Paleocene' etc.; better to pick one convention and stick to it throughout. I think the pre-PETM, PETM and EECO nomenclature is clearer. E.g. page 4, line 2-3 (?); page 9 line 12, page 11 line 8, and others. c. Use the term 'core' instead of alternatives. e.g.: Page 4, line 2(?): change 'four main simulations' to 'four core simulations'. Or, use 'main' instead of 'core' throughout. Page 11, line 11: 'core' instead of 'standard'. Better to check throughout.

2. 'palaeo' and 'paleo' are interchanged throughout. Better to choose one convention and stick to it, since GMD is an EGU journal, I recommend 'palaeo'. Please correct throughout.

3. In sections 4.2.3 and 4.2.5, the choice to use a higher solar constant (1365 W m-2) than what is suggested for the latest Palaeocene-Eocene (1359 W m-2; Gough,

1981; see manuscript) is justified by stating that it will in part counteract using lower atmospheric CH4 than probably existed (and vice versa). I struggle to accept this justification. Using the updated CMIP6 preindustrial solar constant (see point 32) would provide a much smaller difference between the latest Palaeocene-Eocene and present day solar constants (+2 W m-2). Besides this, without a quantified effect of each (solar constant versus CH4), this speculation seems to be very vague, and the effects are likely to be non-linear, surely. Since these are relatively straight forward boundary conditions to implement in the model (compared to palaeogeography, for example), why not use a more suitable solar constant (presumably 1359 W m-2) and a representative CH4 – few of the boundary conditions are certain, but if we know CH4 was elevated then surely it should be in the model set-up. Otherwise what can be achieved by the model-data comparison? This also effects section 4.3.5. It is a valuable sensitivity study, but with regard to my comment on this above, this section might need rethinking/phrasing (e.g. the sensitivity study to use the preindustrial value of 1361 W m-2, or others if the literature presents alternatives to 1359 W m-2/indicates the uncertainty on this).

SPECIFIC COMMENTS: (suggested inserted text in italics)

4. Page 2, line 5: 'Together with the CMIP6 preindustrial simulation, these form the first' (or other such indication that the preindustrial simulation is part of the core experiment; see comment 1)

5. Page 2, line 7: 'core palaeoclimate simulations, one core preindustrial simulation and a set of'

6. Page 2, line 17-18: 'It also aims to assess their relevance for our understanding of future climate change.' This would be a valuable addition, but I don't think it's really followed up later. I suggest adding a brief section to the article explicitly dealing with this.

7. Page 2, line 19: I checked in CMIP and PMIP and I don't think this will be part

of CMIP, so maybe make this a little clearer here; from this line I was left with the impression that DMIP will be in CMIP6.

8. Page 2, line 22 and throughout: proxy for what? Suggest 'climate proxy'. This should be checked throughout and always amended so that it is clear what the 'proxy' is a proxy for.

9. In general there is a misuse of 'which', when used for restrictive clauses it should be 'that', though maybe this is different in American English: a. Page 2, line 25 b. Page 2, line 26 c. Page 3, line 13 d. Page 5, line 3 e. Page 9, line 1(?) f. Page 9, line 22 g. Page 10, line 19 h. Page 11, line 28

10. Page 2, line 26: 'of particular relevance' for what?

11. Page 3, line 2: suggest summarising the intriguing model-data mismatches and inconsistencies between 'proxies'.

12. Page 3, line 3-4: insert commas after 'Gasson et al. (2014)', 'Lunt et al. (2013)' and 'Carmichael et al. (2016)'. Change comma to semi-colon after 'inception' and 'Eocene simulations'.

13. Page 3, line 8: suggest rephrasing 'proxy-proxy differences' (see comment 8. 'data' used previously, or could be more specific: 'differences between geological data').

14. Page 3, line 9-10: suggest reordering the time periods so that they are chronological (and again below in lines 19-21).

15. Page 3, lines 19-21: as well as reordering (comment 14), suggest adding a brief description of these time periods to make it clear what they are and why they were specifically chosen (e.g. a brief description under each numbered list element); otherwise that information is lacking. In particular, this information should explicitly (but not exclusively) tie-back to (i), (ii) and (iii) from lines 11-14; perhaps at least one sentence on each.

16. Page 3, line 23-24: 'The pre-PETM…and the EECO'. I'm sure this is true, but it's not very clear how or why this is true. Addressing comment 15 would probably solve this.

17. Page 3, line 29-30: after 'recent interest in…relevance to future warming' add some example references.

18. Page 4, line 8-9: so would this then constitute 5 core simulations for those groups?

19. Page 4, line 10: add simulation names in header '(pre-PETM, PETM, EECO)'

20. Page 4, line 11: clarify that 'three core palaeoclimate simulations'; there are four (or five – comment 18) core simulations.

21. Section 4.2: It's a little unclear as to what boundary conditions relate to which of the three core palaeoclimate simulations. It would be helpful if this could be clarified through the text in this section.

22. Section 4.2.1: So, are all groups expected to adjust their model's bathymetry in line with the boundary conditions? Can/will all groups do this? If not, maybe add a few lines on this so it's clear.

23. Page 4, line 14: remove back-to-back parentheses, adjust to 'Herold et al. (2014; henceforth H14)'

24. Section 4.2.2 (iv) river runoff: do some models compute this from their orography and land-sea mask?

25. Section 4.2.3: it would be helpful to add a figure compiling and summarising the greenhouse gas concentrations (at least for CO2) over this period from the geological data, including uncertainty. I understand the time axis would probably need to expand over a substantially wider period that these simulations cover, but then the periods represented by the three palaeoclimate simulations could be highlighted (e.g. vertical shaded bars if time is on x-axis). It would give helpful context as well as summarise

the uncertainty. The 1x, 3x, 6x and 12x CO2 values (plus 2x and 4x?) could also be indicated (e.g. dashed horizontal lines).

26. Section 4.2.3: This is entitled 'Greenhouse gas concentrations', but really only addresses CO2. I suggest at least adding a discussion and presentation of CH4 boundary conditions (see comment 3), but otherwise rename this section appropriately.

27. Page 6: line 7-8: add refs for the records showing this (CO2 and extant temperature records). Possibly also clarify what 'extant temperature records' means in this context; is it the temperature proxy archive that survives or the temperature reconstruction?

28. Page 7: some extra commas are needed: Line 5 after '(see Section 4.2.5)' Line 6 after 'In effect'

29. Page 7, line 6: 'at the CMIP6 preindustrial concentrations'?

30. Page 7, line 8: 'terms of global surface temperature'? This is unclear so needs clarifying.

31. Page 7, line 10-11; can this also be justified scientifically? What are the implications/added value of the results of these 2x and 4x CO2 simulations?

32. Page 7, line 27: the solar constant is out of date. The CMIP6 preindustrial value will be 1361.0 W m−2 (Matthes et al., 2016). Also affects page 10, line 23.

33. Page 8, line 6: replace 'SSTs' with 'Sea Surface Temperatures (SSTs)'

34. Page 8, line 24: Do you mean 'hydrological' instead of 'geological'? Otherwise I'm not sure what is meant by 'geological cycling'.

35. Page 9, line 7: what is the address/location/reference for the PMIP database?

36. Page 9, line 7: replace 'in the Appendix' with 'in Appendix 1, including Tables 1-3'.

37. Page 9, line 9: 'Appendix 1, Tables 1-3'.

38. Page 9: some extra commas are needed: line 26: after 'Ideally' line 30: after

'studies'

39. Page 10, lines 4-6: why carry out sensitivity studies of 'widening/constricting and shallowing/deepening key ocean gateways, raising/lowering mountain ranges, and changing the bathymetry of ocean shelves'? Please summarise (from the literature) the kind of changes or uncertainties in these boundary conditions that are thought to have taken place during this period, and what effect they may/may not have had?

40. Page 10, line 27: what should be there instead of 'Section ??'; is it 'Section 4.2.6' or 'Section 4.2.7'? Where is this discussed? I think the discussion needs adding to one of these sections (4.2.6 or 4.2.7 or both).

41. Page 10, line 28: 'will be a function of'.

42. Page 11, line 17: 'will be to develop new ways'.

43. Page 11, line 22: remove parentheses from within parentheses: 'see Dowsett et al., 2012)'.

44. Page 11, line 29: add comma: 'In this respect, we are'

45. Page 11, line 29: reference the PlioMIP special issue properly, because I assume that is why the URL is given (i.e. in addition to the Haywood et al. ref).

46. Page 12, line 8: Change 'Appendix A' to 'Appendix 1' (or vice versa earlier).

47. Page 12, line 9: 'variables below (Tables 1-3) should be submitted'

48. Table 2: replace 'SST' with 'Sea surface temperature', replace 'T' with 'potential temperature' (I assume it is potential temperature?), replace 'S' with 'salinity'.

Reference cited in review: Matthes, K., Funke, B., Anderson, M. E., Barnard, L., Beer, J., Charbonneau, P., Clilverd, M. A., Dudok de Wit, T., Haberreiter, M., Hendry, A., Jackman, C. H., Kretschmar, M., Kruschke, T., Kunze, M., Langematz, U., Marsh, D. R., Maycock, A., Misios, S., Rodger, C. J., Scaife, A. A., Seppälä, A., Shangguan,

M., Sinnhuber, M., Tourpali, K., Usoskin, I., van de Kamp, M., Verronen, P. T. and Versick, S.: Solar Forcing for CMIP6 (v3.1), Geosci. Model Dev. Discuss., 1–82, doi:10.5194/gmd-2016-91, 2016.

---

## Editor Comment (EC3) · J. C. Hargreaves (Editor) · 29 Sep 2016

Sorry to be slow to post this - I hope this doesn't cause you too much inconvenience - I was going away when the paper went into final response and seem to be taking far too long to catch up with things.

I agree with the reviewer's comment that a table summarising the protocols is required, and think there are a couple of other things missing too.

1. To improve comprehension for those not immersed in DeepMIP intervals, I need to see some kind of visual timeline which indicates what the climate was like during these intervals. Then I can see when the intervals were, and have some understanding of what the differences in climate were both between the intervals and relative to the climate throughout the Earth's history.

[Figure]

2. Paleoclimate simulations are meaningless without data and the data section is worryingly fanciful. I want to see actual description of datasets, or if these are being developed as part of the project, then a much clearer timeline of what will be made available when (how many points are expected for what variables etc). If this is presently impossible, then there would be the possibility of writing a companion paper to this one outlining the data sets (from the GMD Manuscript Types page, ".Papers describing data sets designed for the support and evaluation of model simulations are within scope. These data sets may be syntheses of data which have been published elsewhere. The data sets must also be made available, and any code used to create the syntheses should also be made available.").

Finally: GMD is indeed an EGU journal and papers should be in English, but a while ago they changed from requiring British English to allowing whatever flavour of English you prefer. But, as one of the reviewers says, you are supposed to be consistent within the paper.

[Surely it's Palæo ? :-) ]

---

## Author Response (AR1)

This document includes a response to all the Reviewer and Editor comments. This is then followed by a revised version of the manuscript in which all our proposed changes are clearly highlighted (including line numbers which are referenced by this document). We thank both Reviewers and the Editor for their comments.

**Reviewer 1**

It is proposed to run sensitivity experiments to account for different sets of boundary conditions, and experiments with different CO2 levels to account for the uncertainties in the CO2 reconstructions, which is interesting to test climate sensitivity to CO2 under these conditions. With this perspective, the experimental design could be improved to better liaise with the CMIP6 exercise. In particular, the CMIP6 DECK includes a preindustrial and an abrupt4xCO2 experiment, in which the CO2 level is quadrupled from the pre-industrial level. The DeepMIP protocol recommends to run the pre-industrial as in CMIP6 but it would be very interesting for the groups to also run the CMIP6 DECK abrupt4xCO2 simulation. If the DeepMIP protocol also included a similar 4xCO2 experiment with the deep-time continents and ocean, then it would be easy to examine whether the deep time continents and oceans have an impact on the Earth's sensitivity to greenhouse gases increase, and how much can be inferred on climate sensitivity from these climates.

Done. We agree that it would be very interesting to insist that all models carry out a CMIP6-style *abrupt-4xCO2* simulation, so we have added this. See Table 1 and Section 4.1.

I would therefore argue for changing the priorities in the experimental design (cf page 7, lines 9-11) and to test 2x and 4xCO2 (and higher) first, rather than 3x and 6xCO2. The pre-industrial control and the abrupt4xCO2 experiments have also been proposed to be mandatory for modelling groups wishing to take part in the PMIP4 exercise, to better liaise with CMIP6, so the above recommendation would also warrant a better relation to PMIP4 activities.

The value of 6×PI (1680ppmv) is chosen for the EECO because this is in agreement with the value reconstructed by Anagnostou et al (2016) of 1625±760ppmv. We have now made this clearer in the text. Furthermore, the CMIP6 *abrupt-4*×CO2 CMIP6 simulation is an abrupt forcing, and only runs for 150 years, so is not directly comparable with our Eocene simulations anyway. See Page 8, Line 11, and Figure 5.

Apart from the more specific comments below, what is missing from the manuscript at this stage is a table summarizing the experiments and boundary conditions, and the names given to the experiments so that all groups use these names. Additional figures could also be inserted to better illustrate the scope of the project and the different options in boundary conditions, as explained in the comments below.

As suggested, we added a table of experiments (see Table 1). As many of the sensitivity studies are qualitative suggestions, without formal designs, we only include those sensitivity studies which are formally defined ( $CO_2$  and palaeogeography).

pages 2-3, section 2: this section on previous work could be illustrated by a figure showing what can be improved from this previous work.

**Done. See new Figure 1.**

page 3, section 3: for outsiders, it would be good to have a figure locating the three periods in a broader chronology of the Earth climate evolution.

**Done. See new Figure 2.**

page 4, section 4.2, lines 11-12: "There are three standard simulations" seem to contrast with the sentence at the top of the page: "The DeepMIP experimental protocol consists of four main simulations". It would be good to clarify this: three periods, but four simulations.

Done. We now state "The DeepMIP experimental protocol consists of five main simulations (pre-industrial, future, two early Eocene, and one latest Paleocene/pre-PETM), plus a number of optional sensitivity studies (see Section 4.3).". And later... "There are three standard paleoclimate simulations (*deepmip-stand-3xCO2*, *deepmip-stand-6xCO2*, *deepmip-stand-12xCO2*), which differ only in their atmospheric CO2 concentration, plus a number of optional sensitivity studies."

page 4, section 4.2.1: it would be good to stress at this point that the same paleogeography is used for all three periods.

**Done. This is also clear now in the new Table 1.**

Also, the main cautionary points in the implementation of the paleogeography, such as straits and shallow basins, should be highlighted, in relation with the sensitivity experiments proposed in section4.3.2. Done. Added "Care should be taken when defining the land-sea mask for the ocean component of the model that the various seaways are preserved at the model resolution; this may require some manual manipulation of the land-sea mask.". Page 6, Line 24.

page 5, lines 7ff, about the soils: I am not very familiar with this issue, but I would expect spatial heterogeneities in soil properties, so how can these be prescribed "homogeneously"?

Yes, they should be globally constant as there is no robust data on the heterogeneities in soil properties. Clarified: "Parameters associated with soils should be given constant values over the globe, with values for these parameters (e.g. albedo, water-holding capacity etc.) given by the global-mean of the group's pre-industrial simulation.". Page 8, Line 9.

page 7, value of the solar constant: it has been revised to 1361 W/m2 (Matthes et al, http://www.geosci-model-devdiscuss.net/gmd-2016-91/). Since this paper is still in discussion, it will be worth referring to its final value when it is out. However, this has an impact on the discussion about early Eocene values in section 4.2.5 and on the sensitivity experiments proposed in section 4.3.5. Is the value found by Gough (1981) actually tied to a present value of 1365 W/m2?

The original formula in Gough (1981) is relative to the modern value, and not an absolute. Therefore a change in the preindustrial control value also affects the Eocene value. We now state: "The solar constant in the CMIP6 *piControl* simulation is defined as 1361.0Wm-2 (Matthes, in review, 2016). Although the early Eocene (51 Ma) solar constant was ~0.43% less than this (Gough, 1981), i.e. ~1355Wm-2, ....". Page 10, Line 7.

page 7, justification of not changing the solar constant in the DeepMIP experiment, to counteract the absence of elevated CH4 in the design. This should be better justified. Both forcings are not equivalent and it is rather easy to change the CH4 values in the models. At least the radiative forcing from the CH4 high values should be evaluated and compared to the non-changes in the solar constant.

We have made a calculation of the radiative forcing due to the change in solar constant and due to an increase in CH4 from preindustrial values to 3000 ppbv, which is a typical value found by Beerling et al. The radiative forcings are -1.03 W/m2 and +0.98 W/m2 respectively. As such, we do think we are justified in assuming these two forcings will approximately cancel out. Furthermore, it does make the sensitivity analysis of the causes of EECO/PETM warmth compared to modern much simpler. We have added this calculation to the text. Page 9, Line 18.

page 9: sensitivity to paleogeography: maps of differences could be shown to convince modelling groups that it is worth investing the time to perform these sensitivity experiments. **Done. Figure 3 now includes all 3 recommended palaeogeographies.**

A practical question is about where to actually find this other paleogeography. **Table 2 now details where all files are located.**

page 12: the PMIP data base should be used ! this is the only way cross-period analyses can be performed and other groups can be involved, bringing additional diagnostics and analyses. So the list in Table 1 should be expressed in terms of PMIP/CMIP6 variables. In particular, the acronyms "FLNS", "FLNT" etc should be explained.

Changed "Ideally" to "We strongly recommend that". Note that the FLNS and FLNT acronyms are explained in the footnote to the Table. Page 11, Line 21.

page 2, line 7. Replace "paleo simulations" by "paleoclimate simulations" (we hope that the simulations are new, and not "paleo")

Done throughout.

page 2, line 15: "deep-time model intercomparison project". should this be "deep-time climates"? The project does not aim at comparing deep times, but rather their climates, doesn't it?

**We understand the reviewer's comment, but the name of the MIP is already defined, see www.deepmip.org.**

pages 4 and 5: references should be added for the CESM and CLM models. **Done.**

page7, line 25: the Louvain-la-Neuve group has recommended to use the term "astronomical parameters" rather than "orbital parameters" since obliquity is not an element describing the orbit of the Earth.

page 9, line 9: reference to Appendix 1 should be changed to Appendix A. **Done.**

page 10, line 27: the link to the section is missing **Done.**

page 11, last line: parentheses are missing around the web site reference. **Done.**

**Reviewer 2**

In some places, some expansion of the text is required to clearly explain what may already be apparent to experts immersed in the science, but would be helpful information for the less well-versed. These mainly relate to summarising existing literature and would not be fundamental changes to the manuscript structure or protocol details.

**We have added new Figures 1 and 2 which illustrate the context of the various time periods, and the issues around model-data comparison.**

1. There needs to be better consistency between the way the core simulations are referred to:

a. Whether there are 3 or 4 (I understand that there are 3 palaeo simulations and 1 preindustrial simulation and that these are the core, but this is not clear enough in the manuscript when interchanging between describing 3 and 4 core simulations): We now refer consistently to "5 main simulations", "3 standard palaeoclimate simulations", "2 relevant simulations from CMIP6", and "sensitivity studies".

b. How the palaeoclimate simulations are named as both 'pre-PETM', 'PETM' and 'EECO' versus 'two early Eocene, and one latest Paleocene' etc.; better to pick one convention and stick to it throughout. I think the pre-PETM, PETM and EECO nomenclature is clearer. E.g. page 4, line 2-3 (?); page 9 line 12, page 11 line 8, and others.

We are now consistent. When referring to the time periods, we refer to 'early Eocene' or 'latest Paleocene'. When referring to the simulations themselves, we us EECO/PETM/pre-PETM.

c. Use the term 'core' instead of alternatives. e.g.: Page 4, line 2(?): change 'four main simulations' to 'four core simulations'. Or, use 'main' instead of 'core' throughout. Page 11, line 11: 'core' instead of 'standard'. Better to check throughout. We now refer consistently to "5 main simulations", "3 standard palaeoclimate simulations", "2 relevant simulations from CMIP6", and "sensitivity studies".

2. 'palaeo' and 'palaeo' are interchanged throughout. Better to choose one convention and stick to it, since GMD is an EGU journal, I recommend 'palaeo'. Please correct throughout.

**We are now consistent. We use "palaeo" apart from for the official stratigraphic name "Paleocene" and for the official name "Paleoclimate Model Intercomparison Project"**

3. In sections 4.2.3 and 4.2.5, the choice to use a higher solar constant (1365 W m-2) than what is suggested for the latest Palaeocene-Eocene (1359 W m-2; Gough, 1981; see manuscript) is justified by stating that it will in part counteract using lower atmospheric CH4 than probably existed (and vice versa). I struggle to accept this justification. Using the updated CMIP6 preindustrial solar constant (see point 32) would provide a much smaller difference between the latest Palaeocene-Eocene and present day solar constants (+2 W m-2). Besides this, without a quantified effect of each (solar constant versus CH4), this speculation seems to be very vague, and the effects are likely to be non-linear, surely. Since these are relatively straight

forward boundary conditions to implement in the model (compared to palaeogeography, for example), why not use a more suitable solar constant (presumably 1359 W m-2) and a representative CH4 – few of the boundary conditions are certain, but if we know CH4 was elevated then surely it should be in the model set-up. Otherwise what can be achieved by the model-data comparison?

The original formula for solar constant in Gough (1981) is relative to the modern value, and not an absolute. Therefore a change in the preindustrial control value also affects the Eocene value. We now state: "The solar constant in the CMIP6 *piControl* simulation is defined as 1361.0Wm-2 (Matthes, in review, 2016). Although the early Eocene (51 Ma) solar constant was ~0.43% less than this (Gough, 1981), i.e. ~1355Wm-2, ....". Furthermore, we have made a calculation of the radiative forcing due to the change in solar constant and due to an increase in CH4 from preindustrial values to 3000 ppbv, which is a typical value found by Beerling et al. The radiative forcings are -1.03 W/m2 and +0.98 W/m2 respectively. As such, we do think we are justified in assuming these two forcings will approximately cancel out. Furthermore, it does make the sensitivity analysis of the causes of EECO/PETM warmth compared to modern much simpler. We have added this calculation to the text. See Sections 4.2.3 and 4.2.5.

This also effects section 4.3.5. It is a valuable sensitivity study, but with regard to my comment on this above, this section might need rethinking/ phrasing (e.g. the sensitivity study to use the preindustrial value of 1361 W m-2, or others if the literature presents alternatives to 1359 W m-2/indicates the uncertainty on this).

We now clarify that the suggested reduction is 0.43%, which for a modern solar constant of 1361W/m2 becomes 1355.15 W/m2. Page 10, Line 8.

4. Page 2, line 5: 'Together with the CMIP6 preindustrial simulation, these form the first' (or other such indication that the preindustrial simulation is part of the core experiment; see comment 1) **Done.**

5. Page 2, line 7: 'core palaeoclimate simulations, one core preindustrial simulation and a set of' **Done.**

6. Page 2, line 17-18: 'It also aims to assess their relevance for our understanding of future climate change.' This would be a valuable addition, but I don't think it's really followed up later. I suggest adding a brief section to the article explicitly dealing with this.

Added "In particular, we anticipate papers that explore the relevance of the DeepMIP simulations and climate proxy syntheses for future climate, for example through model developments that arise as a result of the model-data comparison, or emergent constraints (Bracegridle and Stephenson, 2013) on global-scale metrics such as climate sensitivity.". Page 15, Line 3.

7. Page 2, line 19: I checked in CMIP and PMIP and I don't think this will be part of CMIP, so maybe make this a little clearer here; from this line I was left with the impression that DMIP will be in CMIP6.

With the new structure of CMIP6, all of PMIP (including DeepMIP) can be considered as being under the umbrella of CMIP6, so we think the current text is correct. Only a limited number of PMIP simulations are "Tier 1" CMIP6 simulations, but all of PMIP is within CMIP6.

8. Page 2, line 22 and throughout: proxy for what? Suggest 'climate proxy'. This should be checked throughout and always amended so that it is clear what the 'proxy' is a proxy for. **Done throughout.**

9. In general there is a misuse of 'which', when used for restrictive clauses it should be 'that', though maybe this is different in American English: a. Page 2, line 25 b. Page 2, line 26 c. Page 3, line 13 d. Page 5, line 3 e. Page 9, line 1(?) f. Page 9, line 22 g. Page 10, line 19 h. Page 11, line 28

After googling this, I realise that I have been writing incorrect English for the last 35 years! Thanks for pointing this out. Now corrected throughout I think.

10. Page 2, line 26: 'of particular relevance' for what?

Added: "This is of particular relevance to models that are also used for future projection" [note use of "that" in relation to comment 9. directly above!]. Page 3, Line 1.

11. Page 3, line 2: suggest summarising the intriguing model-data mismatches and inconsistencies between 'proxies'. We now reference Figure 1 which highlights these issues explicitly. Also added "For example, proxy-derived SST estimates indicate a weak meridional temperature gradient during the early Eocene which cannot easily be reconciled with model simulations". Page 3, Line 8.

12. Page 3, line 3-4: insert commas after 'Gasson et al. (2014)', 'Lunt et al. (2013)' and 'Carmichael et al. (2016)'. Change comma to semi-colon after 'inception' and 'Eocene simulations'. **Done.**

13. Page 3, line 8: suggest rephrasing 'proxy-proxy differences' (see comment 8. 'data' used previously, or could be more specific: 'differences between geological data').

Done. Changed to "and a greater understanding for the reasons behind differences between different climate proxies". Page 3, Line 15.

14. Page 3, line 9-10: suggest reordering the time periods so that they are chronological (and again below in lines 19-21). They are chronological! (from a geologists point of view). We also prefer this way because then we can introduce the PETM acronym before using pre-PETM.

15. Page 3, lines 19-21: as well as reordering (comment 14), suggest adding a brief description of these time periods to make it clear what they are and why they were specifically chosen (e.g. a brief description under each numbered list element); otherwise that information is lacking. In particular, this information should explicitly (but not exclusively) tie-back to (i), (ii) and (iii) from lines 11-14; perhaps at least one sentence on each.

Done. Note that (i),(ii), and (iii) are covered in the subsequent sentences.

16. Page 3, line 23-24: 'The pre-PETM: : :and the EECO'. I'm sure this is true, but it's not very clear how or why this is true. Addressing comment 15 would probably solve this. **Done.**

17. Page 3, line 29-30: after 'recent interest in: : :relevance to future warming' add some example references. Done: "Furthermore, due at least in part to interest in the Eocene and PETM for providing information of relevance to the future (e.g. Anagnostou et al, 2016; Zeebe et al, 2016), there is a relative wealth of climate proxy data with which the model results can be compared.". Page 5, Line 7.

18. Page 4, line 8-9: so would this then constitute 5 core simulations for those groups?
As suggested, we have now changed the naming conventions. We now refer consistently to "5 main simulations", "3 standard palaeoclimate simulations", "2 relevant simulations from CMIP6", and "sensitivity studies".

19. Page 4, line 10: add simulation names in header '(pre-PETM, PETM, EECO)' **Done.**

20. Page 4, line 11: clarify that 'three core palaeoclimate simulations'; there are four (or five – comment 18) core simulations. We now refer consistently to "5 main simulations", "3 standard palaeoclimate simulations", "2 relevant simulations from CMIP6", and "sensitivity studies".

21. Section 4.2: It's a little unclear as to what boundary conditions relate to which of the three core palaeoclimate simulations. It would be helpful if this could be clarified through the text in this section.

**Table 1 clarifies the relationship between the boundary conditions and the simulations.**

22. Section 4.2.1: So, are all groups expected to adjust their model's bathymetry in line with the boundary conditions? Can/will all groups do this? If not, maybe add a few lines on this so it's clear.

All groups should change the bathymetry. Given the large change in land-sea mask, it is hard to imagine groups attempting to change the land-sea mask but not the bathymetry.

23. Page 4, line 14: remove back-to-back parentheses, adjust to 'Herold et al. (2014; henceforth H14)' **Done.**

24. Section 4.2.2 (iv) river runoff: do some models compute this from their orography and land-sea mask? As far as we are aware, most models allow this field to be prescribed. We added the filename and variable to Table 2.

25. Section 4.2.3: it would be helpful to add a figure compiling and summarising the greenhouse gas concentrations (at least for CO2) over this period from the geological data, including uncertainty. I understand the time axis would probably need to expand over a substantially wider period that these simulations cover, but then the periods represented by the three palaeoclimate simulations could be highlighted (e.g. vertical shaded bars if time is on x-axis). It would give helpful context as well as summarise the uncertainty. The 1x, 3x, 6x and 12x CO2 values (plus 2x and 4x?) could also be indicated (e.g. dashed horizontal lines).

**Done – see new Figure 5.**

26. Section 4.2.3: This is entitled 'Greenhouse gas concentrations', but really only addresses CO2. I suggest at least adding a discussion and presentation of CH4 boundary conditions (see comment 3), but otherwise rename this section appropriately. We now discuss CH4 in more detail, and have added an additional sensitivity study to CH4 in the latter sections, especially for those groups who can predict CH4 interactively. See Section 4.2.3 and 4.3.6.

27. Page 6: line 7-8: add refs for the records showing this (CO2 and extant temperature records). Possibly also clarify what 'extant temperature records' means in this context; is it the temperature proxy archive that survives or the temperature reconstruction?

We clarify by citing the benthic oxygen isotope record. This implies that PETM temperatures were similar to EECO temperatures, which implies the CO2 concentrations were also similar. Page 8, Line 7.

28. Page 7: some extra commas are needed: Line 5 after '(see Section 4.2.5)' Line 6 after 'In effect' **Done.**

29. Page 7, line 6: 'at the CMIP6 preindustrial concentrations'? **Done.**

30. Page 7, line 8: 'terms of global surface temperature'? This is unclear so needs clarifying. **We have removed this sentence.**

31. Page 7, line 10-11; can this also be justified scientifically? What are the implications/ added value of the results of these 2x and 4x CO2 simulations?

Added "In this way, the modelled Eocene climate sensitivity and its nonlinearities can be investigated.". Page 9, Line 24.

32. Page 7, line 27: the solar constant is out of date. The CMIP6 preindustrial value will be 1361.0 W m□2 (Matthes et al., 2016). Also affects page 10, line 23.

**Done. Page 10, Line 7. And page 13, Line 19.**

33. Page 8, line 6: replace 'SSTs' with 'Sea Surface Temperatures (SSTs)' **Done.**

34. Page 8, line 24: Do you mean 'hydrological' instead of 'geological'? Otherwise I'm not sure what is meant by 'geological cycling'.

Replaced with "on these timescales long-term geological sources and sinks of NaCl associated with crustal recycling also play an important role;....". page 11, Line 9.

35. Page 9, line 7: what is the address/location/reference for the PMIP database? This has not yet been set up or decided. Added "...uploaded to the anticipated PMIP database". Page 11, Line 22.

36. Page 9, line 7: replace 'in the Appendix' with 'in Appendix 1, including Tables 1-3'. **Done.**

37. Page 9, line 9: 'Appendix 1, Tables 1-3'. **Done.**

38. Page 9: some extra commas are needed: line 26: after 'Ideally' line 30: after 'studies' **Done.**

39. Page 10, lines 4-6: why carry out sensitivity studies of 'widening/constricting and shallowing/deepening key ocean gateways, raising/lowering mountain ranges, and changing the bathymetry of ocean shelves'? Please summarise (from the literature) the kind of changes or uncertainties in these boundary conditions that are thought to have taken place during this period, and what effect they may/may not have had?

Added "The exact geometry and state of these features are not all well constrained geologically; therefore it is interesting to explore the uncertainties in climate which may result from uncertainties in their configuration.". here we also provide some more justification for these sensitivity studies (see Author Changes section below). See paragraph beginning Page 12, Line 23.

40. Page 10, line 27: what should be there instead of 'Section ??'; is it 'Section 4.2.6' or 'Section 4.2.7'? Where is this discussed? I think the discussion needs adding to one of these sections (4.2.6 or 4.2.7 or both). **Done.**

41. Page 10, line 28: 'will be a function of'. **Done.**

42. Page 11, line 17: 'will be to develop new ways'. **Done.**

43. Page 11, line 22: remove parentheses from within parentheses: 'see Dowsett et al., 2012)'. **Done.**

44. Page 11, line 29: add comma: 'In this respect, we are' **Done.**

45. Page 11, line 29: reference the PlioMIP special issue properly, because I assume that is why the URL is given (i.e. in addition to the Haywood et al. ref).

Done.

46. Page 12, line 8: Change 'Appendix A' to 'Appendix 1' (or vice versa earlier). **Done.**

47. Page 12, line 9: 'variables below (Tables 1-3) should be submitted' **Done.**

48. Table 2: replace 'SST' with 'Sea surface temperature', replace 'T' with 'potential temperature' (I assume it is potential temperature?), replace 'S' with 'salinity'.

**Done.**

**Editor comments**

1. To improve comprehension for those not immersed in DeepMIP intervals, I need to see some kind of visual timeline which indicates what the climate was like during these intervals. Then I can see when the intervals were, and have some understanding of what the differences in climate were both between the intervals and relative to the climate throughout the Earth's history.

Done. See new Figure 2.

2. Paleoclimate simulations are meaningless without data and the data section is worryingly fanciful. I want to see actual description of datasets, or if these are being developed as part of the project, then a much clearer timeline of what will be made available when (how many points are expected for what variables etc). If this is presently impossible, then there would be the possibility of writing a companion paper to this one outlining the data sets (from the GMD Manuscript Types page, ".Papers describing data sets designed for the support and evaluation of model simulations are within scope. These data sets may be syntheses of data which have been published elsewhere. The data sets must also be made available, and any code used to create the syntheses should also be made available.").

Yes, we do intend to write a paper summarising the vision for these datasets, and have already embarked on this process. This may well end up being a companion GMD paper.

Finally: GMD is indeed an EGU journal and papers should be in English, but a while ago they changed from requiring British English to allowing whatever flavour of English you prefer. But, as one of the reviewers says, you are supposed to be consistent within the paper. [Surely it's Palæo ? :-) ]

We are now consistent. We use "palaeo" apart from for the official stratigraphic name "Paleocene" and for the official name "Paleoclimate Model Intercomparison Project".

**Author Changes**

We have added 5°C to our recommended initial temperature state for the ocean. This is to likely shorten the timescale of equilibration of the simulations. See Equation 1.

We have expanded the justification for the paleogeographic sensitivity studies. See paragraph beginning Page 12, Line 23.

We have made a number of additional minor spelling and grammatical changes

We added the following co-authors because they have contributed to the paper and/or DeepMIP: Jeff Kiehl, Eleni Anagnostou, Aradhna Tripati, Gordon Inglis, Stephen Jones, and Henk Dijkstra.

**DeepMIP: experimental design for model simulations of the EECO, PETM, and pre-PETM.**

Daniel J. Lunt1, Matthew Huber2, Eleni Anagnostou9, Michiel L.J. Baatsen3, Rodrigo Caballero4, Rob DeConto5, Henk A. Dijkstra3, Yannick Donnadieu6, David Evans31, Ran Feng8, Gavin Foster9, Ed Gasson5, Anna S. von der Heydt3, Chris J. Hollis10, Gordon N. Inglis32, Stephen M. Jones34, Jeff Kiehl29, Sandy Kirtland Turner11, Robert L. Korty12, Reinhardt Kozdon13, Srinath Krishnan7, Jean-Baptiste Ladant6, Petra Langebroek14, Caroline H. Lear15, Allegra N. LeGrande16, Kate Littler17, Paul Markwick18, Bette Otto-Bliesner8, Paul Pearson15, Christopher J. Poulsen19, Ulrich Salzmann20, Christine Shields8, Kathryn Snell21, Michael Starz22, James Super7, Clay Tabor8, Jess Tierney23, Gregory J.L. Tourte1, Aradhna Tripati33, Gary R. Upchurch24, Bridget S. Wade25, Scott L. Wing26, Arne M.E. Winguth27, Nicky Wright28, James C. Zachos29, and Richard Zeebe30

10GNS Science, New Zealand

16NASA-GISS, USA

17Camborne School of Mines, University of Exeter, UK

18Getech Group plc, UK

- 19Department of Earth and Environmental Sciences, University of Michigan, USA
- 20Department of Geography, Northumbria University, UK
- 21Department of Geological Sciences, University of Colorado, USA
- 22Alfred Wegener Institute, Germany
- 23Department of Geosciences, University of Arizona, USA
- 24Department of Biology, Texas State Univesity, USA
- 25Department of Earth Sciences, University College London, UK
- 26Department of Paleobiology, Smithsonian Institution, USA

27Earth and Environmental Science, University of Texas - Arlington, USA

28School of Geosciences, University of Sydney, Australia

- 30Department of Oceanography, University of Hawaii at Manoa, USA
- 31Earth Sciences, University of St Andrews, UK
- 32School of Chemistry, University of Bristol, UK

<sup>1School of Geographical Sciences, University of Bristol, UK

<sup>2Department of Earth Sciences, University of New Hampshire, USA

<sup>3Institute for Marine and Atmospheric research Utrecht (IMAU), Utrecht University, Netherlands

<sup>4Department of Meteorology (MISU), Stockholm University, Sweden

<sup>5Department of Geosciences, University of Massachusetts-Amherst ,USA

<sup>6Laboratoire des Sciences du Climat et de l'Environnement, CNRS/CEA, France

<sup>7Department of Geology and Geophysics, Yale University, USA

<sup>8National Centre for Atmospheric Research, USA

[revised manuscript text omitted]

deepmip-stand-X×CO2
deepmip-stand-X×CO2 | Topography
Vegetation
Runoff | Supp Info of H14
Supp Info of H14
Supp Info of H14 | herold_etal_eocene_topo_1x1.nc
herold_etal_eocene_biome_1x1.nc
herold_etal_eocene_runoff_1x1.nc | topo
eocene_biome [3]
RTM_FLOW_DIRECT |
| deepmip-sens-geoggetech                                                          | Topography                         | Supp Info of Lunt et al. (2016)                          | bath_ypr.nc, orog_ypr.nc                                                                              | bathuk, oroguk                                         |
| deepmip-sens-geogpalmag                                                          | Topography                         | Supp Info of this paper                                  | Herold2014_TPW.nc                                                                                     | Band1                                                  |

[1] Where *X* can be 3,6, or 12.

[3] 27 biomes. For simplified 11 biomes, use variable eocene\_biome-hp.

---

## Author Response (AR2)

**All page and line numbers refer to the "latexdiff" version of the manuscript, appended to this response, which is a comparison of this submission with our previous submission.**

**In response to comments from Astrid Kerweg, we have modified the title to: "The DeepMIP contribution to PMIP4: experimental design for model simulations of the EECO, PETM, and pre-PETM (version 1.0)". This is consistent with other PMIP4 papers currently in review in GMDD (e.g. Otto-Bliesner et al). We add the version number in recognition that this can be useful if changes are needed at a later date (e.g. Schmidt et al, 2011; Schmidt et al, 2012). See Page 1.**

**We realised that we did not include the palaeogeographical rotation files in Supp Info, as stated in the text. These have now been added.**

**We noticed that the Herold et al Supp info does not include land-sea mask. As such we now define how the land-sea masks should be calculated.**
**See Page 6, lines 12, 13, 22, 23.**

Since the reviewer thought it was misleading, I checked with CMIP6, and for the sentence explaining the relationship between DEEPMIP, PMIP and CMIP, please write the following, "DeepMIP is a working group in the wider Paleoclimate Modelling Intercomparison Project (PMIP4), which itself is a part of the sixth phase of the Coupled Model Intercomparison Project (CMIP6, Eyring et al., 2016)." Please include Eyring et al in the reference list.
**Done. See Page 2, lines 19, 20.**

For the data availability, please can you clarify the level of public access for the PMIP and DeepMIP databases? If it is not to be entirely public, how may potential users gain access?
**Added in the Data Availability section: "Data held in both the CMIP6 and DeepMIP Model databases, when these are operational, will likely be freely accessible through data portals after registration.". See Page 15.**

Please also clarify in the data availability section what it is that is contained in table 2. Does this cover all the boundary conditions for all the other experiments?
**Done. Yes, all the sensitivity studies in DeepMIP can be carried out using the files listed in Table 2. Caption of Table 2 modified, see Page 7.**

With regard to this, "We strongly recommend that all model output should be provided in CMIP6-compliant netcdf format, including the standard PMIP variables, and uploaded to the anticipated PMIP database. However, if this is not possible, then netcdf files of the variables in Appendix A, including Tables 3-5, should be uploaded to the DeepMIP Modelling Database, which will be set up if and when required." ... as I read it, there are two things here that are being discussed as if they were one thing. One is the ability to produce CMIP6-compliant netcdf files, and the second is the ability to produce all the PMIP standard variables. I think these two issues should be discussed separately so that it is clearly stated under what conditions the DeepMIP database should be used.
**We have modified this sentence to simply state: "As stated in Section 4.2.8, "We strongly recommend that model output should be uploaded to the CMIP6 database. However, if the CMIP6 database cannot be used, the variables in Tables 3-5 should be uploaded to DeepMIP Model Database, which will be set up if and when required.". See Page 15.**
**Also we have modified the section on Output format: "We strongly recommend that DeepMIP model output should be uploaded to the anticipated PMIP4 component of the CMIP6 database (Eyring et al, 2016), distributed through the Earth System Grid Federation (ESGF).". See Page 11, lines 14-16.**

I am a bit confused by the addition of more authors at this stage, when the revision has mostly consisted of improving the explanation and context. I think it would be a appropriate to add a paragraph outlining the author contributions.
**We have modified the Author Contribution section to state: "A first draft of this paper was written by Dan Lunt and Matt Huber. It was subsequently edited based on discussions at a DeepMIP meeting in January 2016 at NCAR, Boulder, Colorado, USA, and following further**

**email discussions with the DeepMIP community. All authors contributed at the meeting and/or in the subsequent email discussions.". See page 15.**

Otto-Bliesner, B. L., Braconnot, P., Harrison, S. P., Lunt, D. J., Abe-Ouchi, A., Albani, S., Bartlein, P. J., Capron, E., Carlson, A. E., Dutton, A., Fischer, H., Goelzer, H., Govin, A., Haywood, A., Joos, F., Legrande, A. N., Lipscomb, W. H., Lohmann, G., Mahowald, N., Nehrbass-Ahles, C., Pausata, F. S.-R., Peterschmitt, J.-Y., Phipps, S., and Renssen, H.: The PMIP4 contribution to CMIP6 – Part 2: Two Interglacials, Scientific Objective and Experimental Design for Holocene and Last Interglacial Simulations, Geosci. Model Dev. Discuss., doi:10.5194/gmd-2016-279, in review, 2016.

Schmidt, G. A., Jungclaus, J. H., Ammann, C. M., Bard, E., Braconnot, P., Crowley, T. J., Delaygue, G., Joos, F., Krivova, N. A., Muscheler, R., Otto-Bliesner, B. L., Pongratz, J., Shindell, D. T., Solanki, S. K., Steinhilber, F., and Vieira, L. E. A.: Climate forcing reconstructions for use in PMIP simulations of the last millennium (v1.0), Geosci. Model Dev., 4, 33-45, 2011.

Schmidt, G. A., Jungclaus, J. H., Ammann, C. M., Bard, E., Braconnot, P., Crowley, T. J., Delaygue, G., Joos, F., Krivova, N. A., Muscheler, R., Otto-Bliesner, B. L., Pongratz, J., Shindell, D. T., Solanki, S. K., Steinhilber, F., and Vieira, L. E. A.: Climate forcing reconstructions for use in PMIP simulations of the Last Millennium (v1.1), Geosci. Model Dev., 5, 185-191, 2012.

[revised manuscript text omitted]